# Statistical and Geometrical Properties of Regularized Kernel Kullback-Leibler Divergence

**Clémentine Chazal**
CREST, ENSAE, IP Paris
clementine.chazal@ensae.fr

**Anna Korba**
CREST, ENSAE, IP Paris
anna.korba@ensae.fr

**Francis Bach**
INRIA - Ecole Normale Supérieure
PSL Research university
francis.bach@inria.fr

## Abstract

In this paper, we study the statistical and geometrical properties of the Kullback-Leibler divergence with kernel covariance operators (KKL) introduced by Bach [2022]. Unlike the classical Kullback-Leibler (KL) divergence that involves density ratios, the KKL compares probability distributions through covariance operators (embeddings) in a reproducible kernel Hilbert space (RKHS), and compute the Kullback-Leibler quantum divergence. This novel divergence hence shares parallel but different aspects with both the standard Kullback-Leibler between probability distributions and kernel embeddings metrics such as the maximum mean discrepancy. A limitation faced with the original KKL divergence is its inability to be defined for distributions with disjoint supports. To solve this problem, we propose in this paper a regularized variant that guarantees that the divergence is well defined for all distributions. We derive bounds that quantify the deviation of the regularized KKL to the original one, as well as finite-sample bounds. In addition, we provide a closed-form expression for the regularized KKL, specifically applicable when the distributions consist of finite sets of points, which makes it implementable. Furthermore, we derive a Wasserstein gradient descent scheme of the KKL divergence in the case of discrete distributions, and study empirically its properties to transport a set of points to a target distribution.

## 1 Introduction

A fundamental task in machine learning is to approximate a target distribution $q$. For example, in Bayesian inference [Gelman et al., 1995], it is of interest to approximate posterior distributions of the parameters of a statistical model for predictive inference. This has led to the vast development of parametric methods from variational inference [Blei et al., 2017], or non-parametric ones such as Markov Chain Monte Carlo (MCMC) [Roberts and Rosenthal, 2004], and more recently particle-based optimization [Liu and Wang, 2016, Korba et al., 2021]. In generative modelling [Brock et al., 2019, Ho et al., 2020, Song et al., 2020, Franceschi et al., 2023] only samples from $q$ are available and the goal is to generate data whose distribution is similar to the training set distribution. Generally, this problem can be cast as an optimization problem over $\mathcal{P}(\mathbb{R}^d)$, the space of probability distributions over $\mathbb{R}^d$, where the optimization objective is chosen as a dissimilarity function $\mathcal{D}(\cdot|q)$ (a distance or divergence) between probability distributions, that only vanishes at $q$. Starting from an initial distribution $p_0$, a descent scheme can then be applied such that the trajectory $(p_t)_{t \geq 0}$ approaches $q$. In particular, on the space of probability distributions with bounded second moment $\mathcal{P}_2(\mathbb{R}^d)$, one

38th Conference on Neural Information Processing Systems (NeurIPS 2024).

can consider the Wasserstein gradient flow of the functional $\mathcal{F}(p) = \mathcal{D}(p||q)$. The latter defines a path of distributions, ruled by a velocity field, that is of steepest descent for $\mathcal{F}$ with respect to the Wasserstein-2 distance from optimal transport.

This approach has led to a large variety of algorithms based on the choice of a specific dissimilarity functional $\mathcal{F}$, often determined by the information available on the target $q$. For example, in Bayesian or variational inference, where the target's density is known up to an intractable normalizing constant, a common choice for the cost is the Kullback-Leibler (KL) divergence, whose optimization is tractable in that setting [Wibisono, 2018, Ranganath et al., 2014]. When only samples of $q$ are available, it is not convenient to choose the optimization cost as the KL, as it is only defined for probability distributions $p$ that are absolutely continuous with respect to $q$. In contrast, it is more convenient to choose an $\mathcal{F}$ that can be written as integrals against $q$, for instance, maximum mean discrepancy (MMD) [Arbel et al., 2019, Hertrich et al., 2024b], sliced-Wasserstein distance [Liutkus et al., 2019] or Sinkhorn divergence [Genevay et al., 2018]. However, sliced-Wasserstein distances, that average optimal transport distances of 1-dimensional projections of probability distributions (slices) over an infinite number of directions, have to be approximated by a finite number of directions in practice [Tanguy et al., 2023]; and Sinkhorn divergences involve solving a relaxed optimal transport problem. In contrast, MMD can be written in closed-form for discrete measures thanks to the reproducing property of positive definite kernels. The MMD represents probability distributions through their kernel mean embeddings in a reproducing kernel Hilbert space (RKHS), and compute the RKHS norm of the difference of embeddings (namely, the witness function). Moreover, the MMD flow with a smooth kernel (e.g., Gaussian) as in Arbel et al. [2019] is easy to implement, as the velocity field is expressed as the gradient of the witness function, and preserve discrete measures. However, due to the non-convexity of the MMD in the underlying Wasserstein geometry [Arbel et al., 2019], its gradient flow is often stuck in local minimas in practice even for simple target as Gaussian $q$, calling for adaptive schemes tuning the level of noise or kernel hyperparameters [Xu et al., 2022, Galashov et al., 2024], or regularizing the kernel [Chen et al., 2024]. MMD with non-smooth kernels, e.g., based on negative distances [Sejdinovic et al., 2013], have also attracted attention recently, as their gradient flow enjoys better empirical convergence properties than the previous ones [Hertrich et al., 2024a,b]. However, their gradient flow does not preserve discrete measures; and their practical simulation rely on implicit time discretizations [Hertrich et al., 2024a] or slicing [Hertrich et al., 2024b].

In contrast to the MMD with smooth kernels, the KL divergence is displacement convex [Villani, 2009, Definition 16.5] when the target is log-concave (i.e., $q$ has a density $q \propto e^{-V}$ with $V$ convex), and its gradient flow enjoys fast convergence when $q$ satisfies a log-Sobolev inequality [Bakry et al., 2014]. In this regard, it enjoys better geometrical properties than the MMD. Moreover, the KL divergence is equal to infinity for singular $p$ and $q$, which makes its gradient flow extremely sensitive to mismatch of support, so that the flow enforces the concentration on the support of $q$ as desired. On the downside, while the Wasserstein gradient flow of KL divergences is well-defined [Chewi et al., 2020], its associated particle-based discretization is difficult to simulate when only samples of $q$ are available, and a surrogate optimization problem usually needs to be introduced [Gao et al., 2019, Ansari et al., 2020, Simons et al., 2022, Birrell et al., 2022a, Liu et al., 2022]. However, it is unclear whether this surrogate optimization problem preserves the geometry of the KL flow.

Recently, Bach [2022] introduced alternative divergences based on quantum divergences evaluated through kernel covariance operators, that we call here a kernel Kullback-Leibler (KKL) divergence. The latter can be seen as second-order embeddings of probability distributions, in contrast with first-order kernel mean embeddings (as used in MMD). In Bach [2022], it was shown that the KKL enjoys nice properties such as separation of measures, and that it is framed between a standard KL divergence (from above) and a smoothed KL divergence (from below), i.e., a KL divergence between smoothed versions of the measures with respect to a specific smoothing kernel. Hence, it cannot directly be identified to a KL divergence and corresponds to a novel and distinct divergence. However, many of its properties remained unexplored, including a complete analysis of the KKL for empirical measures, a tractable closed-form expression and its optimization properties. In this paper, we tackle the previous questions. We propose a regularized version of the KKL that is well-defined for any discrete measures, in contrast with the original KKL. We establish upper bounds that quantify the deviation of the regularized KKL to its unregularized counterpart, and convergence for empirical distributions. Moreover, we derive a tractable closed-form for the regularized KKL and its derivatives that writes with respect to kernel Gram matrices, leading to a practical optimization algorithm. Finally,

we investigate empirically the statistical properties of the regularized KKL, as well as its geometrical properties when using it as an objective to target a probability distribution $q$.

This paper is organized as follows. Section 2 introduces the necessary background and the regularized KKL. Section 3 presents our theoretical results on the deviation and finite-sample properties of the latter. Section 4 provides the closed-form of regularized KKL for discrete measures as well as the practical optimization scheme based on an explicit time-discretisation of its Wasserstein gradient flow. Section 5 discusses closely related work including distances or divergences between distributions based on reproducing kernels. Finally, Section 6 illustrates the statistical and optimization properties of the KKL on a variety of experiments.

## 2   Regularized kernel Kullback-Leibler (KKL) divergence

In this section, we state our notations and previous results on the (original) kernel Kullback-Leibler (KKL) divergence introduced by Bach [2022], before introducing our proposed regularized version.

**Notations.**   Let $\mathcal{P}(\mathbb{R}^d)$ the set of probability measures on $\mathbb{R}^d$. Let $\mathcal{P}_2(\mathbb{R}^d)$ the set of probability measures on $\mathbb{R}^d$ with finite second moment, which becomes a metric space when equipped with Wasserstein-2 ($W_2$) distance [Villani, 2009].

For $p \in \mathcal{P}(\mathbb{R}^d)$, we denote that $p$ is absolutely continuous w.r.t. $q$ using $p \ll q$, and we use $dp/dq$ to denote the Radon-Nikodym derivative. We recall the standard definition of the Kullback-Leibler divergence, $\mathrm{KL}(p||q) = \int \log(dp/dq)dp$ if $p \ll q$, $+\infty$ else.

If $g : \mathbb{R}^d \to \mathbb{R}^r$ is differentiable, we denote by $Jg : \mathbb{R}^d \to \mathbb{R}^{r \times d}$ its Jacobian. If $r = 1$, we denote by $\nabla g$ the gradient of $g$ and $\mathbf{H}g$ its Hessian. If $r = d$, $\nabla \cdot g$ denotes the divergence of $g$, i.e., the trace of the Jacobian. We also denote by $\Delta g$ the Laplacian of $g$, where $\Delta g = \nabla \cdot \nabla g$. We also denote $I$ the identity matrix or operator.

For a positive semi-definite kernel $k : \mathbb{R}^d \times \mathbb{R}^d \to \mathbb{R}$, its RKHS $\mathcal{H}$ is a Hilbert space with inner product $\langle \cdot, \cdot \rangle_{\mathcal{H}}$ and norm $\| \cdot \|_{\mathcal{H}}$. For $q \in \mathcal{P}_2(\mathbb{R}^d)$ such that $\int k(x,x)dq(x) < \infty$, the inclusion operator $\iota_q : \mathcal{H} \to L^2(q)$, $f \mapsto f$ is a bounded operator with its adjoint being $\iota_q^* : L^2(q) \to \mathcal{H}$, $f \mapsto \int k(x, \cdot)f(x)dq(x)$ [Steinwart and Christmann, 2008, Theorem 4.26 and 4.27]. The covariance operator w.r.t. $q$ is defined as $\Sigma_q = \int k(\cdot, x) \otimes k(\cdot, x)dq(x) = \iota_q^* \iota_q$, where $(a \otimes b)c = \langle b, c \rangle_{\mathcal{H}} a$ for $a, b, c \in \mathcal{H}$. It can also be written $\Sigma_q = \int_{\mathbb{R}^d} \varphi(x)\varphi(x)^* dq(x)$ where $*$ denotes the transposition in $\mathcal{H}$ (recall that for $u \in \mathcal{H}$, $uu^* : \mathcal{H} \to \mathcal{H}$ denotes the operator $uu^*(f) = \langle f, u \rangle_{\mathcal{H}}u$ for any $f \in \mathcal{H}$).

**Kernel Kullback-Leibler divergence (KKL).**   For $p, q \in \mathcal{P}(\mathbb{R}^d)$, the kernel Kullback-Leibler divergence (KKL) is defined in Bach [2022] as:

$$\mathrm{KKL}(p||q) := \mathrm{Tr}(\Sigma_p \log \Sigma_p) - \mathrm{Tr}(\Sigma_p \log \Sigma_q) = \sum_{(\lambda,\gamma) \in \Lambda_p \times \Lambda_q} \lambda \log\left(\frac{\lambda}{\gamma}\right) \langle f_\lambda, g_\gamma \rangle_{\mathcal{H}}^2. \quad (1)$$

where $\Lambda_p$ and $\Lambda_q$ are the set of eigenvalues of the covariance operators $\Sigma_p$ and $\Sigma_q$, with associated eigenvectors $(f_\lambda)_{\lambda \in \Lambda_p}$ and $(g_\gamma)_{\gamma \in \Lambda_q}$. The KKL (1) evaluates the Kullback-Leibler divergence between operators on Hilbert Spaces, that is well-defined for any couple of positive Hermitian operators with finite trace, at the operators $\Sigma_p$ and $\Sigma_q$. From Bach [2022, Proposition 4], if $p$ and $q$ are supported on compact subset of $\mathbb{R}^d$, and if $k$ is a continuous positive definite kernel with $k(x, x) = 1$ for all $x \in \mathbb{R}^d$, and if $k^2$ is universal [Steinwart and Christmann, 2008, Definition 4.52], then $\mathrm{KKL}(p||q) = 0$ if and only if $p = q$. In Bach [2022], it also was proven that the KKL is upper bounded by the (standard) KL-divergence between probability distributions (see Proposition 4 therein) and lower bounded by the same KL but evaluated at smoothed versions of the distributions, where the smoothing is a convolution with respect to a specific kernel (see Section 4 therein). Thus, the KKL defines a novel divergence between probability measures. It defines then an interesting candidate as to compare probability distributions, for instance when used as an optimization objective over $\mathcal{P}(\mathbb{R}^d)$, in order to approximate a target distribution $q$.

**Definition of the regularized KKL.**   A major issue that the KKL shares with the standard Kullback-Leibler divergence between probability distributions, is that it diverges if the support of $p$ is not

included in the one of $q$ (1). Indeed, for the $\mathrm{KKL}(p||q)$ to be finite, we need $\mathrm{Ker}(\Sigma_q) \subset \mathrm{Ker}(\Sigma_p)$. This condition is satisfied when the support of $p$ is included in the support of $q$. Indeed, if $f \in \mathrm{Ker}(\Sigma_q)$, then $\langle f, \Sigma_q f \rangle_{\mathcal{H}} = \int_{\mathbb{R}^d} f(x)^2 dq(x) = 0$, and so $f$ is zero on the support of $q$, then also on the support of $p$. Hence, the $\mathrm{KKL}$ is not a convenient discrepancy when $q$ is a discrete measure (in particular, if $p$ is also discrete with different support than $q$). A simple fix that we propose in this paper is to consider a regularized version of KKL which is, for $\alpha \in ]0, 1[$,

$$\mathrm{KKL}_\alpha(p||q) := \mathrm{KKL}(p||(1-\alpha)q + \alpha p) = \mathrm{Tr}(\Sigma_p \log \Sigma_p) - \mathrm{Tr}(\Sigma_p \log((1-\alpha)\Sigma_q + \alpha\Sigma_p)). \quad (2)$$

The advantage of this definition is that $\mathrm{KKL}_\alpha$ is finite for any distribution $p, q$. It smoothes the distribution $q$ by mixing it with the distribution $p$, to a degree determined by the parameter $\alpha$. This divergence approximates the original KKL divergence without requiring the distribution $p$ to be absolutely continuous with respect to $q$ for finiteness. Moreover, for any $\alpha \in ]0, 1[$, $\mathrm{KKL}_\alpha(p||q) = 0$ if and only if $p = q$. As $\alpha \to 0$, we recover the original KKL (1), and as $\alpha \to 1$, this quantity converges pointwise to zero.

**Remark 1.** The regularization we consider in (2) has also been considered for the standard KL divergence [Lee, 2000]. These objects, as well as their symmetrized version, were also referred to in the literature as skewed divergences [Kimura and Hino, 2021]. The most famous one is Jensen-Shannon divergence, recovered as a symmetrized skewed KL divergence for $\alpha = \frac{1}{2}$, that is defined as $\mathrm{JS}(p||q) = \mathrm{KL}(p||\frac{1}{2}p + \frac{1}{2}q) + \mathrm{KL}(q||\frac{1}{2}p + \frac{1}{2}q)$.

## 3 Skewness and concentration of the regularized KKL

In this section we study the skewness of the regularized KKL due to the introduction of the parameter $\alpha$, as well as its concentration properties for empirical measures.

**Skewness.** We will first analyze how the regularized KKL behaves with respect to the regularization parameter $\alpha$. First, we show it is monotone with respect to $\alpha$ in the following Proposition.

**Proposition 2.** *Let $p \ll q$. The function $\alpha \mapsto \mathrm{KKL}_\alpha(p||q)$ is decreasing on $[0, 1]$.*

Proposition 2 shows that the regularized KKL shares a similar monotony behavior than the regularized, or skewed, (standard) KL between probability distributions, as recalled in Appendix A.1. The proof of Proposition 2 can be found in Appendix B.1. It relies on the positivity of the KKL divergence, and the use of the identity [Ando, 1979]

$$\mathrm{Tr}(\Sigma_p(\log \Sigma_p - \log \Sigma_q)) = \int_0^{+\infty} \mathrm{Tr}\big(\Sigma_p(\Sigma_p + \beta I)^{-1}\big) - \mathrm{Tr}\big(\Sigma_q(\Sigma_q + \beta I)^{-1}\big)d\beta, \quad (3)$$

where $I$ is the identity operator, that is used in all our proofs. We now fix $\alpha \in ]0, 1[$ and provide a quantitative result about the deviation of the regularized (or skewed) KKL to its original counterpart.

**Proposition 3.** *Let $p, q \in \mathcal{P}(\mathbb{R}^d)$. Assume that $p \ll q$ and that $\frac{dp}{dq} \leqslant \frac{1}{\mu}$ for some $\mu > 0$. Then,*

$$|\mathrm{KKL}_\alpha(p||q) - \mathrm{KKL}(p||q)| \leqslant \left(\alpha\left(1 + \frac{1}{\mu}\right) + \frac{\alpha^2}{1-\alpha}\left(1 + \frac{1}{\mu^2}\right)\right) |\mathrm{Tr}(\Sigma_p \log \Sigma_q)|. \quad (4)$$

Proposition 3 recovers a similar quantitative bound than the one we can obtain for the standard KL between probability distributions, see Appendix A.2; and state that the skewness of the regularized KKL can be controlled by the regularization parameter $\alpha$, especially when the latter is small. However, the tools used to derive this inequality are completely different by nature than for the KL case. Its complete proof can be found in Appendix B.2, but we provide here a sketch.

*Sketch of proof.* Let $\Gamma = \alpha\Sigma_p + (1-\alpha)\Sigma_q$. We write $\mathrm{KKL}(p||q)_\alpha - \mathrm{KKL}(p||q) = \mathrm{Tr}\,\Sigma_p \log \Sigma_q - \mathrm{Tr}\,\Sigma_p \log \Gamma$ that we write as (3). In order to upper bound this integral we use the operator equalities, for two operators $A$ and $B$, $A^{-1} - B^{-1} = A^{-1}(B - A)B^{-1} = A^{-1}(B - A)A^{-1} - A^{-1}(B - A)B^{-1}(B - A)A^{-1}$ which we apply to $A = \Gamma + \beta I$ and $B = \Sigma_q + \beta I$. We then use the assumption $\mu\Sigma_p \preccurlyeq \Sigma_q$ and carefully apply upper bounds on positive semi-definite operators, using the matrix inequality results from Appendix A.3, to conclude the proof. $\square$

**Statistical properties.** We now focus on the regularized KKL for empirical measures and derive finite-sample guarantees.

**Proposition 4.** *Let $p, q \in \mathcal{P}(\mathbb{R}^d)$. Assume that $p \ll q$ with $\frac{dp}{dq} \leqslant \frac{1}{\mu}$ for some $0 < \mu \leqslant 1$ and let $\alpha \leqslant \frac{1}{2}$. We remind that $\varphi(x)$ is the feature map of $x \in \mathbb{R}^d$ in the RKHS $\mathcal{H}$. Assume also that $c = \int_0^{+\infty} \sup_{x \in \mathbb{R}^d} \langle \varphi(x), (\Sigma_p + \beta I)^{-1} \varphi(x) \rangle_{\mathcal{H}}^2 d\beta$ is finite. Let $\hat{p}, \hat{q}$ supported on $n$, $m$ i.i.d. samples from $p$ and $q$ respectively. We have:*

$$\mathbb{E}|\mathrm{KKL}_\alpha(\hat{p}||\hat{q}) - \mathrm{KKL}_\alpha(p||q)| \leqslant \frac{35}{\sqrt{m \wedge n}} \frac{1}{\alpha \mu}(2\sqrt{c} + \log n)$$

$$+ \frac{1}{m \wedge n}\left(1 + \frac{1}{\mu} + \frac{c(24 \log n)^2}{\alpha \mu^2}(1 + \frac{n}{m \wedge m})\right). \quad (5)$$

**Remark 5.** It is possible to calculate a similar bound for the above proposition which does not require the condition $p \ll q$. This bound, which can be found at the end of Appendix B.3.3, worsens as $\alpha$ approaches 0 because it scales in $O(\frac{1}{\alpha^2})$ instead of $O(\frac{1}{\alpha})$ above.

The latter proposition extends significantly Bach [2022, Proposition 7] that provided an upper bound on the entropy term only, i.e., the first term in (1):

$$\mathbb{E}[|\mathrm{Tr}(\Sigma_{\hat{p}} \log \Sigma_{\hat{p}}) - \mathrm{Tr}(\Sigma_p \log \Sigma_p)|] \leqslant \frac{1 + c(8 \log n)^2}{n} + \frac{17}{\sqrt{n}}(2\sqrt{c} + \log n). \quad (6)$$

Our bound (5) is explicit in the number of samples $n, m$ for $\hat{p}, \hat{q}$, and for $n = m$ we recover similar terms as (6). Our contribution is to upper bound the cross term, i.e., the second term in (1), involving both $p$ and $q$. We do so by closely follow the proof of [Bach, 2022, Proposition 7] in order to bound the cross terms difference. In consequence, our proof involves technical intermediate results, among which concentration of sums of random self-adjoint operators, and estimation of degrees of freedom. The proof of Proposition 4 can be found in Appendix B.3, but we provide here a sketch.

*Sketch of proof.* We denote $\hat{\Gamma} = \alpha \Sigma_{\hat{p}} + (1 - \alpha)\Sigma_{\hat{q}}$ and $\Gamma$ its population counterpart. In order to bound the cross term we write $\mathrm{Tr}\,\Sigma_{\hat{p}} \log \hat{\Gamma} - \mathrm{Tr}\,\Sigma_p \log \Gamma$ using (3). We split the integrals in three terms, with respect to two parameters $0 < \beta_0 < \beta_1$ that we introduce: (a) one for $\beta$ between 0 and $\beta_0$, (b) one for $\beta$ between $\beta_1$ and infinity and (c) an intermediate one. The $\beta_0$ quantity is chosen to be dependent of $m$ and $n$, so that it converge to zero as $n$ and $m$ go to infinity. This way, for (a) the integral between 0 and $\beta_0$ we simply have to bound $\mathrm{Tr}\,\Sigma_p(\Gamma + \beta I)^{-1}$ and $\mathrm{Tr}\,\Sigma_{\hat{p}}(\hat{\Gamma} + \beta I)^{-1}$ by constant or integrable quantities close to 0. Then, for (b), $\beta_1$ is chosen so that it goes to infinity when $n$ and $m$ go to infinity and (b) is bounded by $1/\beta_1$. Finally we upper bound finely enough (c) to compensate for the fact that the bounds of the integrals tend towards 0 and infinity. $\square$

## 4 Time-discretized regularized KKL gradient flow

In this section, we show that the regularized KKL can be implemented in closed-form for discrete measures, as well as its Wasserstein gradient, making its optimization tractable.

**regularized KKL closed-form.** We first describe how to compute the regularized KKL for (any, not necessarily empirical) discrete measures in practice. This will be useful for the practical implementation of regularized KKL optimization coming next. We provide a closed-form for the latter, involving kernel Gram matrices between supports of the discrete measures.

**Proposition 6.** *Let $\hat{p} = \frac{1}{n} \sum_{i=1}^n \delta_{x_i}$ and $\hat{q} = \frac{1}{m} \sum_{j=1}^m \delta_{y_j}$ two discrete distributions. Define $K_{\hat{p}} = (k(x_i, x_j))_{i,j=1}^n \in \mathbb{R}^{n \times n}$, $K_{\hat{q}} = (k(y_i, y_j))_{i,j=1}^m \in \mathbb{R}^{m \times m}$, $K_{\hat{p},\hat{q}} = (k(x_i, y_j))_{i,j=1}^{n,m} \in \mathbb{R}^{n \times m}$. Then, for any $\alpha \in ]0, 1[$, we have:*

$$\mathrm{KKL}_\alpha(\hat{p}||\hat{q}) = \mathrm{Tr}\left(\frac{1}{n}K_{\hat{p}} \log \frac{1}{n}K_{\hat{p}}\right) - \mathrm{Tr}(I_\alpha K \log(K)),$$

$$\text{where } I_\alpha = \begin{pmatrix} \frac{1}{\alpha}I & 0 \\ 0 & 0 \end{pmatrix} \text{ and } K = \begin{pmatrix} \frac{\alpha}{n}K_{\hat{p}} & \sqrt{\frac{\alpha(1-\alpha)}{nm}}K_{\hat{p},\hat{q}} \\ \sqrt{\frac{\alpha(1-\alpha)}{nm}}K_{\hat{q},\hat{p}} & \frac{1-\alpha}{m}K_{\hat{q}} \end{pmatrix}. \quad (7)$$

Proposition 6 extends non-trivially the result of Bach [2022, Proposition 6] that only provided a closed-form for the entropy term $\mathrm{Tr}(\Sigma_{\hat{p}} \log(\Sigma_{\hat{p}}))$, that corresponds to our first term in Equation (7). Its complete proof can be found in Appendix B.4 but we provide here a sketch.

*Sketch of proof.* Our goal there is to derive a closed-form for the cross-term in $\hat{p}, \hat{q}$ of the KKL, that is $\mathrm{Tr}(\Sigma_{\hat{p}} \log(\alpha\Sigma_{\hat{p}} + (1-\alpha)\Sigma_{\hat{q}}))$. It is based on the observation that if we define $\phi_x = (\varphi(x_1), \ldots, \varphi(x_n))^*$, $\phi_y = (\varphi(y_1), \ldots, \varphi(y_m))^*$ and $\psi$ the concatenation of $\sqrt{\frac{\alpha}{n}}\phi_x$ and $\sqrt{\frac{1-\alpha}{m}}\phi_y$, then the covariance operators write $\Sigma_{\hat{p}} = \psi^T I_\alpha \psi$ and $\alpha\Sigma_{\hat{p}} + (1-\alpha)\Sigma_{\hat{q}} = \psi^T \psi$. Then, the matrices $K_{\hat{p}}$ and $K$ write $K_{\hat{p}} = \psi I_\alpha \psi^T$ and $\psi\psi^T = K$. Finally, we apply an intermediate result (Lemma 12) to obtain the expression of $\log(\alpha\Sigma_{\hat{p}} + (1-\alpha)\Sigma_{\hat{q}})$ as a function of $\log K$. □

**Gradient flow and closed-form for the derivatives.** We now discuss how to optimize $p \mapsto \mathrm{KKL}_\alpha(p\|q)$ for a given target distribution $q$. For a given functional $\mathcal{F}: \mathcal{P}_2(\mathbb{R}^d) \to \mathbb{R}^+$, a Wasserstein gradient flow of $\mathcal{F}$ can be thought as an analog object to a Euclidean gradient flow in the metric space $(\mathcal{P}_2(\mathbb{R}^d), W_2)$ [Santambrogio, 2017], which defines a trajectory $(p_t)_{t \geq 0}$ in $\mathcal{P}_2(\mathbb{R}^d)$ following the steepest descent for $\mathcal{F}$ with respect to the $W_2$ distance. It can be characterized by a *continuity equation*:

$$\partial_t p_t + \nabla \cdot (p_t \nabla \mathcal{F}'(p_t)) = 0, \tag{8}$$

where $\mathcal{F}'(p): \mathbb{R}^d \to \mathbb{R}$ is the first variation of $\mathcal{F}$ at $p \in \mathcal{P}_2(\mathbb{R}^d)$. We recall that the first variation at $p \in \mathcal{P}_2(\mathbb{R}^d)$ as defined in Ambrosio et al. [2005, Lemma 10.4.1] is defined, if it exists, as the function $\mathcal{F}': \mathbb{R}^d \to \mathbb{R}$ such that

$$\lim_{\epsilon \to 0} \frac{1}{\epsilon} \mathcal{F}(p + \epsilon\xi) - \mathcal{F}(p) = \int \mathcal{F}'(p)(x)d\xi(x), \tag{9}$$

for any $\xi = q - p$, $q \in \mathcal{P}_2(\mathbb{R}^d)$. To optimize $\mathrm{KKL}_\alpha$, it is then natural to consider its Wasserstein gradient flow and discretize it in time and space. Since $\mathrm{KKL}_\alpha$ is well-defined for discrete measures $\hat{p}, \hat{q}$, we directly derive its first variation for this setting. Our next result yields a closed-form for the first variation of the regularized KKL.

**Proposition 7.** *Consider $\hat{p}, \hat{q}$ and the matrices $K_{\hat{p}}$, $K$ as defined in Proposition 6. Let $g(x) = \frac{\log x}{x}$. Then, the first variation of $\mathcal{F} = \mathrm{KKL}_\alpha(\cdot\|\hat{q})$ at $\hat{p}$ is, for any $x \in \mathbb{R}^d$:*

$$\mathcal{F}'(\hat{p})(x) = 1 + S(x)^T g(K_{\hat{p}})S(x) - T(x)^T g(K)T(x) - T(x)^T A T(x), \tag{10}$$

*where*

$$S(x) = \left(\frac{1}{\sqrt{n}}k(x, x_1), \ldots, \frac{1}{\sqrt{n}}k(x, x_n)\right), \quad T(x) = \left(\sqrt{\frac{\alpha}{n}}k(x, x_1), \ldots; \sqrt{\frac{1-\alpha}{m}}k(x, y_1), \ldots\right),$$

*and* $A = \sum_{j=1}^{n+m} \frac{\|\boldsymbol{a}_j\|^2}{\eta_j} \boldsymbol{c}_j \boldsymbol{c}_j^T + \sum_{j \neq k} \frac{\log \eta_j - \log \eta_k}{\eta_j - \eta_k} \langle \boldsymbol{a}_j, \boldsymbol{a}_k \rangle \boldsymbol{c}_j \boldsymbol{c}_k^T,$

*where $(\boldsymbol{c}_j)_j$ are the eigenvectors of $K$, and $(\boldsymbol{a}_j)_j$ the vectors of first $n$ terms of $(\boldsymbol{c}_j)_j$.*

The proof of Proposition 7 can be found in Appendix B.5, we provide a sketch below.

*Sketch of proof.* Our proof deals separately with the entropy and the cross term, writing $\mathcal{F} = \mathcal{F}_1 + \mathcal{F}_2$. Starting from the definition (9), we denote $\Delta = \varepsilon\Sigma_\xi$. For $\mathcal{F}_1$, we write $\mathcal{F}_1(\hat{p} + \varepsilon\xi) - \mathcal{F}_1(\hat{p}) = \sum_{i=1}^n f(\lambda_i(\Sigma_{\hat{p}} + \Delta)) - f(\lambda_i(\Sigma_{\hat{p}}))$. To write this term, we use the residual formula, which can be used to differentiate eigenvalues of functions. Indeed, we can write, for an operator $A$ with finite number of positive eigenvalues, $\sum_{\lambda \in \Lambda(A)} f(\lambda) = \oint_\gamma f(z) \mathrm{Tr}((zI - A)^{-1})dz$ where $\gamma$ is a loop in $\mathbb{C}$ surrounding all the positive eigenvalues of $A$. Applying this to our case, if we choose $\gamma$ such that it surrounds both the eigenvalues of $\Sigma_{\hat{p}}$ and of $\Sigma_{\hat{p}} + \Delta$, we obtain $\sum_{i=1}^n f(\lambda_i(\Sigma_{\hat{p}} + \Delta)) - f(\lambda_i(\Sigma_{\hat{p}})) = \frac{1}{2i\pi}\oint_\gamma f(z)\mathrm{Tr}((zI - \Sigma_{\hat{p}} - \Delta)^{-1}) - f(z)\mathrm{Tr}((zI - \Sigma_{\hat{p}})^{-1})dz$. Using the identity $A^{-1} - B^{-1} = A^{-1}(B - A)A^{-1} + o(B - A)$, the previous quantity becomes $\sum_{i=1}^n f(\lambda_i(\Sigma_{\hat{p}} + \Delta)) - f(\lambda_i(\Sigma_{\hat{p}})) = \frac{1}{2i\pi}\oint_\gamma \sum_{k=1}^n \frac{f(z)}{(z-\lambda_k)^2}dz\,\mathrm{Tr}(f_k^* f_k \Delta) + o(\varepsilon)$. Under the integral we recognise a holomorphic function with isolated singularities and we can therefore apply the residue formula again. Concerning $\mathcal{F}_2$, we proceed in the same way, with the difference that as we have a cross term, eigenvectors will appear in the calculation and in the final result. □

Leveraging the analytical form for the first variation given by Proposition 7, the Wasserstein gradient of $\mathcal{F} = \mathrm{KKL}_\alpha(\cdot\|\hat{q})$ at $p$ is given by $\nabla\mathcal{F}'(p) : \mathbb{R}^d \to \mathbb{R}^d$ by taking the gradient with respect to $x$ in Equation (10). Notice that the latter only involves derivatives with respect to the kernel $k$, and can be computed in $\mathcal{O}((n+m)^3)$ due to the singular value decomposition of the matrix $K$ defined in Proposition 6.

Starting from some initial distribution $p_0$, and for some given step-size $\gamma > 0$, a forward (or explicit) time-discretization of (8) corresponds to the Wasserstein gradient descent algorithm, and can be written at each discrete time iteration $l \geq 1$ as:

$$p_{l+1} = (\mathrm{Id} - \gamma\nabla\mathcal{F}'(p_l))_{\#}p_l \tag{11}$$

where $\mathrm{Id}$ is the identity map in $L^2(p_l)$ and $\#$ denotes the pushforward operation. For discrete measures $\mu_n = 1/n \sum_{i=1}^n \delta_{x^i}$, we can define $F(X^n) := \mathcal{F}(p_n)$ where $X^n = (x^1, \ldots, x^n)$, since the functional $\mathcal{F}$ is well defined for discrete measures. The Wasserstein gradient flow of $\mathcal{F}$ (8) becomes the standard Euclidean gradient flow of the particle based function $F$. Furthermore, Wasserstein gradient descent (11) writes as Euclidean gradient descent on the position of the particles.

## 5    Related work

**Divergences based on kernels embeddings.**    Kernels have been used extensively to design useful distances or divergences between probability distributions, as they provide several ways to represent probability distributions, e.g., through their kernel mean or covariance embeddings. The Maximum Mean Discrepancy (MMD) [Gretton et al., 2012] is maybe the most famous one. It is defined as the RKHS norm of the difference between the mean embeddings $m_p := \int k(x,\cdot)dp(x)$ and $m_q := \int k(x,\cdot)dq(x)$, i.e., $\mathrm{MMD}(p\|q) = \|m_p - m_q\|_{\mathcal{H}}$. When $k$ is characteristic, $\mathrm{MMD}(p\|q) = 0$ if and only if $p = q$ [Sriperumbudur et al., 2010]. MMD belongs to the family of integral probability metrics [Müller, 1997] as it can be written as $\mathrm{MMD}(p\|q) = \sup_{f\in\mathcal{H}, \|f\|_{\mathcal{H}}\leq 1} \mathbb{E}_p[f(X)] - \mathbb{E}_q[f(X)]$. Alternatively, it can be seen as an $L^2$-distance between kernel density estimators. It became popular in statistics and machine learning through its applications in two-sample test [Gretton et al., 2012], or more recently in generative modeling [Bińkowski et al., 2018].

However, kernel mean embeddings are not the only way (and maybe not the most expressive) to represent probability distributions. For instance, MMD may not be discriminative enough when the distributions differ only in their higher-order moments but have the same mean embedding. For this reason, several works have resorted to test statistics that incorporate the kernel covariance operator of the probability distributions. For instance, Harchaoui et al. [2007] construct a test statistic that resembles and regularizes the $\mathrm{MMD}(p\|q)$ by incorporating covariance operators (more precisely, $\|(\Sigma_{\frac{p+q}{2}} + \beta I)^{-1}(m_p - m_q)\|_{\mathcal{H}}$) yielding in some sense a chi-square divergence between the two distributions. This work has been recently generalized in Hagrass et al. [2022] to more general spectral regularizations, and in Chen et al. [2024] with a different covariance operator. A similar regularized MMD statistic is employed by Balasubramanian et al. [2021], Hagrass et al. [2023] in the context of the goodness-of-fit test.

**Kernel variational approximation of the KL.**    An alternative use of kernels to compute probability divergences is through approximation of variational formulations of $f$-divergences [Nguyen et al., 2010, Birrell et al., 2022b] of which KL-divergence is an example. Indeed, the KL divergence between $p$ and $q$ writes $\sup_{g\in M_b} \int g\,dp - \int e^g dq$ where $M_b$ denotes the set of all bounded measurable functions on $\mathbb{R}^d$. For instance, Glaser et al. [2021] consider a variational formulation of the KL divergence restricted to RKHS functions, namely the KALE divergence:

$$\mathrm{KALE}(p\|q) = (1+\lambda)\max_{g\in\mathcal{H}} \int g\,dp - \int e^g dq - \frac{\lambda}{2}\|g\|_{\mathcal{H}}^2. \tag{12}$$

Recently, Neumayer et al. [2024] extended the latter work and studied kernelized variational formulation of general $f$-divergences, referred to as Moreau envelopes of f-divergences in RKHS, including the KALE as a particular case. They prove that these functionals are lower semi-continuous, and that their Wasserstein gradient flows are well defined for smooth kernels (i.e., the functionals are $\lambda$-convex, and the subdifferential contains a single element). However, the KALE does not have a closed form expression (in constrast to the kernelized variational formulation of chi-square, which writes as a

regularized MMD, see [Chen et al., 2024]). For discrete distributions $p$ and $q$ supported on $n$ atoms, the KALE divergence can be written a strongly convex $n$-dimensional problem, and can be solved using standard Euclidean optimization methods. Still, this makes the simulation of KALE Wasserstein gradient flow (e.g., gradient descent on the positions of particles) computationally demanding, as it requires solving an inner optimization problem at each iteration. This inner optimization problem is solved calling another optimization algorithm. Glaser et al. [2021] use various methods in their experiments, including Newton's method (that scales as $\mathcal{O}(n^3)$ due to the matrix inversion), or less computationally demanding ones such as gradient descent (GD) or coordinate descent. For large values of the regularization parameter $\lambda$, using plain GD works reasonably well, but for small values of $\lambda$, the problem becomes quite ill-conditioned and GD needs to be run with smaller step-sizes. Moreover, as KALE (and its gradient) are not available in closed-form, they cannot be used with fast and hyperparameter-free methods, such as L-BFGS [Liu and Nocedal, 1989] which requires exact gradients. This contrasts with our regularized KKL divergence and its gradient, which are available in closed-form. In our experiments, we will investigate further the relative performance of KALE and KKL.

## 6 Experiments

In this section, we illustrate the validity of our theoretical results and the performance of gradient descent for the regularized KKL. In all our experiments, we consider Gaussian kernels $k(x,y) = \exp\left(-\frac{\|x-y\|^2}{\sigma^2}\right)$ where $\sigma$ denotes the bandwith. Our code is available on the github repository https://github.com/clementinechazal/KKL-divergence-gradient-flows.git.

**Illustrations of skewness and concentration of the** KKL. We first illustrate our results of Proposition 3 and Proposition 4, i.e. the skewness and concentration properties of $\mathrm{KKL}_\alpha$. We investigate these properties for various settings of $p, q$ two fixed probability distributions on $\mathbb{R}^d$, varying the choice of $\alpha$, dimension $d$, and distributions $p, q$. We consider empirical measures $\hat{p}, \hat{q}$ supported on $n$ i.i.d. samples of $p, q$ (in this section we take the same number of samples for both distributions, i.e., $n = m$ in the notations of Section 3), and we observe the concentration of $\mathrm{KKL}_\alpha(\hat{p}, \hat{q})$ around its population limit as the number of samples $n$ (particles) go to infinity.

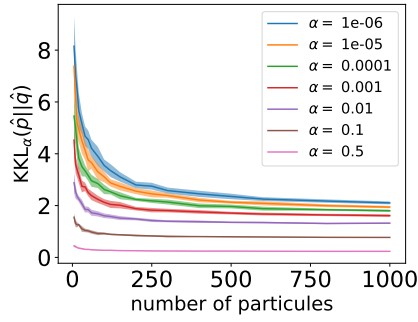

Figure 1: Concentration of empirical $\mathrm{KKL}_\alpha$ for $d = 10$, $\sigma = 10$, $p, q$ Gaussians.

Each time, we plot the results obtained over 50 runs, randomizing the samples drawn from each distribution. Thick lines represent the average value over these multiple runs. We represent the dependence in $\alpha$ and $n$ in dimension 10 in Figure 1, for $p, q$ anisotropic Gaussian distributions with different means and variances. Alternative settings and additional results are deferred to the Appendix C, such as different distributions (e.g. a Gaussian $p$ versus an Exponential $q$), as well as the dimension dependence for a fixed $\alpha$. We can clearly see in Figure 1 the monotony of $\mathrm{KKL}_\alpha$ with respect to $\alpha$ (as stated in Proposition 2) and the concentration of the empirical $\mathrm{KKL}_\alpha$ around its population version, which happens faster for a larger value of $\alpha$, as predicted by our finite-sample bounds in Proposition 4.

**Sampling with** KKL **gradient descent.** Finally, we study the performance of KKL gradient descent in practice, as described in Section 4. We consider two settings already used by Glaser et al. [2021] for KALE gradient flow, reflecting different topological properties for the source-target pair: a pair with a target supported on a hypersurface (zero volume support) and a pair with disjoint supports of positive volume. Alternative settings, e.g. Gaussians source and mixture of Gaussians target that are pairs of distributions with a positive density supported on $\mathbb{R}^d$, are deferred to Appendix C. We also report there additional plots related to the experiments of this section.

We have treated $\alpha$ as a hyperparameter here, and in this section for simplicity of notations we refer to KKL as the objective functional. As both KKL and its gradient can be explicitly computed, one can

implement descent either using a constant step-size, or through a quasi-Newton algorithm such as L-BFGS [Liu and Nocedal, 1989]. The latter is often faster and more robust than the conventional gradient descent and does not require choosing critical hyper-parameters, such as a learning rate, since L-BFGS performs a line-search to find suitable step-sizes. It only requires a tolerance parameter on the norm of the gradient, which is in practice set to machine precision. In contrast, as said in the previous section, the KALE and its gradient are not available in closed-form.

The first example is a target distribution $q$ supported (and uniformly distributed) on a lower-dimensional surface that defines three non-overlapping rings, see Figure 2. The initial source is a Gaussian distribution with a mean in the vicinity of the target $q$. We compare Wasserstein gradient descent of KKL, Maximum Mean Discrepancy [Arbel et al., 2019] and KALE [Glaser et al., 2021], using the code provided in these references. For each method, we choose a bandwith $\sigma = 0.1$, and we optimize the step-size for each method, and sample $n = 100$ points from the source and target distribution. Our method is robust to the choice of $\alpha$ and generally performs very well on this example, as shown in Figure 2. We can notice that since MMD is not sensitive to the difference of support between $p$ and $q$, the particles may leave the rings; while for the regularized KKL flow, as for KALE flow, the particles follow closely the support of the target distribution.

The second example consists of a source and target pair $p, q$ that are supported on disjoint subsets, each with a finite, positive volume, in contrast with the previous example. The source and the target are uniform supported on a heart and a spiral respectively. We again run MMD, KALE and KKL gradient descent. In this example, both KKL and KALE recover the spiral shape, much before the MMD flow trajectory; but both have a harder time recovering outliers, disconnected from the main support of the spiral.

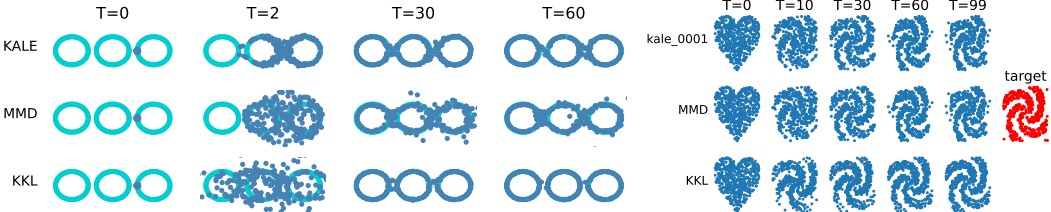

Figure 2: MMD, KALE and KKL flow for 3 rings target.       Figure 3: Shape transfer

# 7 Conclusion

In this work, we investigated the properties of the recently introduced Kernel Kullback-Leibler (KKL) divergence as a tool for comparing probability distributions. We provided several theoretical results, among which quantitative bounds on the deviation from the regularized KKL to the original one, and finite-sample guarantees for empirical measures, that are validated by our numerical experiments. We also derived a closed-form and computable expression for the regularized KKL as well as its derivatives, enabling to implement (Wasserstein) gradient descent for this objective. Our experiments validate the use of KKL as a tool to compare discrete measures, as its gradient flow is much better behaved than the one of Maximum Mean Discrepancy which relies only on mean (first moments) embeddings of probability distributions. It can also be computed in closed-form, in contrast to the KALE divergence introduced recently in the literature, and can benefit from fast and hyperparameter-free methods such as L-BFGS.

While our study has advanced our understanding of the KKL divergence, several limitations must be acknowledged. Firstly, theoretical guarantees for the convergence of the KKL flow remain unestablished. Secondly, reducing the computational cost is crucial for practical applications. Investigating the use of random features presents a promising avenue for making the computations more efficient.

# 8 Acknowledgments

A. Korba and C. Chazal acknowledge the support of ANR PEPR PDE-AI.

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

## A    Additional Background

### A.1    Monotonicity of the $\mathrm{KL}_\alpha$ divergence

**Proposition 8.** *The function $\alpha \mapsto \mathrm{KL}(p||\alpha p + (1 - \alpha)q)$ is decreasing on $[0, 1]$.*

*Proof.* Let $0 < \alpha' < \alpha < 1$,

$$\mathrm{KL}(p||\alpha p + (1 - \alpha)q) - \mathrm{KL}(p||\alpha' p + (1 - \alpha')q)$$

$$= \int (\log(q + \alpha'(p - q)) - \log(q + \alpha(p - q)))dp$$

$$= \int (\log(q + \alpha'(p - q)) - \log(q + \alpha(p - q)))(\alpha dp + (1 - \alpha)dq)$$

$$+ (1 - \alpha) \int (\log(q + \alpha'(p - q)) - \log(q + \alpha(p - q)))(dp - dq)$$

We have $\int (\log(q + \alpha'(p - q)) - \log(q + \alpha(p - q)))(\alpha dp + (1 - \alpha)dq) = -\mathrm{KL}(\alpha p + (1 - \alpha)q)||\alpha' p + (1 - \alpha')q) \leqslant 0$. For the second term, note that because of the increasing nature of the log, for the points for which $p - q > 0$, $\log(q + \alpha'(p - q)) \leqslant \log(q + \alpha(p - q))$ and vice versa. Hence

$$(1 - \alpha) \int (\log(q + \alpha'(p - q)) - \log(q + \alpha(p - q)))(dp - dq) \leqslant 0.$$

This concludes the proof. $\qquad\qquad\square$

### A.2    Skewness of the $\mathrm{KL}_\alpha$ divergence

**Proposition 9.** *Suppose that $dp/dq \leqslant \frac{1}{\mu}$,*

$$\left| \mathrm{KL}(p||\alpha p + (1 - \alpha)q) - \mathrm{KL}(p||q) \right| \leqslant \left( \alpha \left( 1 + \frac{1}{\mu} \right) + \frac{\alpha^2}{1 - \alpha} \left( 1 + \frac{1}{\mu^2} \right) \right) \int \log q \, dp$$

*Proof.* First, we have

$$\left| \mathrm{KL}(p||\alpha p + (1 - \alpha)q) - \mathrm{KL}(p||q) \right| \leq \int \left| (\log q - \log(q + \alpha(p - q))) \right| dp.$$

Now, we remind the following identity which is the real-valued analog of Equation (3). Let $x, y > 0$,

$$\log x - \log y = \int_0^\infty \left( \frac{1}{y + \beta} - \frac{1}{x + \beta} \right) d\beta.$$

Hence,

$$\log q - \log(q + \alpha(p - q)) = \int_0^{+\infty} \left( \frac{1}{q + \alpha(p - q) + \beta} - \frac{1}{q + \beta} \right) d\beta$$

$$= \int_0^{+\infty} \left( \frac{(q + \beta)^2}{(q + \beta)^2(q + \alpha(p - q) + \beta)} - \frac{(q + \alpha(p - q) + \beta)(q + \beta)}{(q + \beta)^2(q + \alpha(p - q) + \beta)} \right) d\beta$$

$$= \int_0^{+\infty} \left( \frac{-\alpha(p - q)(q + \beta)}{(q + \beta)^2(q + \alpha(p - q) + \beta)} \right) d\beta$$

$$= \int_0^{+\infty} \left( \frac{-\alpha^2(p - q)^2 - \alpha(p - q)(q + \alpha(p - q) + \beta)}{(q + \beta)^2(q + \alpha(p - q) + \beta)} \right) d\beta$$

$$= \alpha^2 \int_0^{+\infty} \left( \frac{p - q}{q + \beta} \right)^2 \frac{1}{q + \alpha(p - q) + \beta} d\beta - \alpha \int_0^{+\infty} \frac{p - q}{(q + \beta)^2} d\beta.$$

$$\tag{13}$$

The first term in (13) can be bounded as

$$\left| \alpha \int_0^{+\infty} \frac{p-q}{(q+\beta)^2} d\beta \right| \le \alpha \int_0^{+\infty} \frac{p}{(q+\beta)^2} d\beta + \alpha \int_0^{+\infty} \frac{q}{(q+\beta)^2} d\beta$$

$$\le \alpha \int_0^{+\infty} \frac{1}{\mu(q+\beta)} d\beta + \alpha \int_0^{+\infty} \frac{1}{(q+\beta)} d\beta$$

$$\le \alpha \left( \frac{1}{\mu} + 1 \right) \log q.$$

where the penultimate inequality uses $q \geqslant \mu p$ for the first term. The second term in (13) can be bounded similarly as:

$$\alpha^2 \int_0^{+\infty} \left( \frac{p-q}{q+\beta} \right)^2 \frac{1}{q+\beta+\alpha(p-q)} d\beta \leqslant \alpha^2 \int_0^{+\infty} \frac{p^2+q^2}{(q+\beta)^2} \frac{1}{q+\beta+\alpha(p-q)} d\beta$$

$$\leqslant \frac{\alpha^2}{1-\alpha} \left( \frac{1}{\mu^2} + 1 \right) \log q.$$

Finally,

$$| \mathrm{KL}(p||\alpha p + (1-\alpha)q) - \mathrm{KL}(p||q)| \leqslant \left( \alpha \left( 1 + \frac{1}{\mu} \right) + \frac{\alpha^2}{1-\alpha} \left( 1 + \frac{1}{\mu^2} \right) \right) \int \log q \, dp. \quad \square$$

### A.3 Background operator monotony

We recall here results about matrix and operator monotony, that we extensively use in all our proofs. These are set out in the blog post `https://francisbach.com/matrix-monotony-and-convexity/`, see [Bhatia, 2009] for a more complete reference. For 2 operators $A$ and $B$ in $\mathcal{H}$, we denote $A \preccurlyeq B$ the operators inequality in the sense : $\forall x \in \mathcal{H}$, $x^* A x \leqslant x^* B x$. Let $S$ being the set of symmetric operators and $S^+$ the set of symmetric positive operators. We have

   i) If $A, B \in S$, $X$ another operator in $\mathcal{H}$, $A \preccurlyeq B \Rightarrow X^* A X \preccurlyeq X^* B X$.

   ii) If $A, B \in S$, $M \succcurlyeq 0$, $A \preccurlyeq B \Rightarrow \mathrm{Tr}(AM) \leqslant \mathrm{Tr}(BM)$.

   iii) If $B$ is invertible, $A \preccurlyeq B \Rightarrow B^{-1/2} A B^{-1/2} \preccurlyeq I$.

   iv) If $B \in S$, $B^* B \preccurlyeq I \Rightarrow B B^* \preccurlyeq I$.

   v) If $A, B \in S^+$, $A \preccurlyeq B \Rightarrow A^{1/2} \preccurlyeq B^{1/2}$.

   vi) If $A, B \in S^+$ and are invertible, $A \preccurlyeq B \Rightarrow B^{-1} \preccurlyeq A^{-1}$.

## B  Proofs

### B.1  Proof of Proposition 2

Let $0 < \alpha' < \alpha$. We have:

$\mathrm{KKL}_\alpha(p||q) - \mathrm{KKL}_{\alpha'}(p||q)$
$= \mathrm{Tr}\,\Sigma_p \log\left(\Sigma_q + \alpha'(\Sigma_p - \Sigma_q)\right) - \mathrm{Tr}\,\Sigma_p \log\left(\Sigma_q + \alpha(\Sigma_p - \Sigma_q)\right)$
$= \mathrm{Tr}\left(\alpha\Sigma_p + (1-\alpha)\Sigma_q\right) \log\left(\Sigma_q + \alpha'(\Sigma_p - \Sigma_q)\right) - \mathrm{Tr}\left(\alpha\Sigma_p + (1-\alpha)\Sigma_q\right) \log\left(\Sigma_q + \alpha(\Sigma_p - \Sigma_q)\right)$
$\quad + (1-\alpha)\,\mathrm{Tr}(\Sigma_p - \Sigma_q)\left[\log(\Sigma_q + \alpha'(\Sigma_p - \Sigma_q)) - \log(\Sigma_q + \alpha(\Sigma_p - \Sigma_q))\right].$ (14)

For the first term in (14), we recognize

$$\mathrm{Tr}(\alpha\Sigma_p + (1-\alpha)\Sigma_q) \log(\Sigma_q + \alpha'(\Sigma_p - \Sigma_q)) - \mathrm{Tr}(\alpha\Sigma_p + (1-\alpha)\Sigma_q) \log(\Sigma_q + \alpha(\Sigma_p - \Sigma_q))$$

$$= -\mathrm{KKL}(\alpha p + (1-\alpha)q || \alpha' p + (1-\alpha')q) \leqslant 0.$$

For the second term in (14), we write

$$(1-\alpha)\operatorname{Tr}(\Sigma_p - \Sigma_q)\left[\log(\Sigma_q + \alpha'(\Sigma_p - \Sigma_q)) - \log(\Sigma_q + \alpha(\Sigma_p - \Sigma_q))\right]$$

$$=(1-\alpha)\int_0^{+\infty}\operatorname{Tr}(\Sigma_p - \Sigma_q)\left((\Sigma_q + \beta I + \alpha(\Sigma_p - \Sigma_q))^{-1} - (\Sigma_q + \beta I + \alpha'(\Sigma_p - \Sigma_q))^{-1}\right)d\beta$$

$$=(1-\alpha)(\alpha' - \alpha)\times$$

$$\int_0^{+\infty}\operatorname{Tr}(\Sigma_p - \Sigma_q)(\Sigma_q + \beta I + \alpha(\Sigma_p - \Sigma_q))^{-1}(\Sigma_p - \Sigma_q)(\Sigma_q + \beta I + \alpha'(\Sigma_p - \Sigma_q))^{-1}d\beta,$$

where the last equality uses $A^{-1} - B^{-1} = A^{-1}(B - A)B^{-1}$. The term under the integral writes as $(AX)^T AX$, where $A = (\Sigma_q + \beta I + \alpha(\Sigma_p - \Sigma_q))^{-1}$ and $X = \Sigma_p - \Sigma_q$, hence it is positive. Then, knowing that $(1-\alpha)(\alpha' - \alpha) \leqslant 0$, we conclude that the second term in (14) is negative. Finally,

$$\operatorname{KKL}_\alpha(p||q) - \operatorname{KKL}_{\alpha'}(p||q) \leqslant 0.$$

## B.2 Proof of Proposition 3

This proof makes repeated use of the results about matrix monotony, some of which we recall in Appendix A.3. The reader may refer to Appendix A.2 for analog computations in the KL case. We denote $\Gamma = \alpha\Sigma_p + (1-\alpha)\Sigma_q = \Sigma_q + \alpha(\Sigma_p - \Sigma_q)$. We write using direct integration [Ando, 1979]

$$\operatorname{KKL}(p||q) - \operatorname{KKL}_\alpha(p||q) = \operatorname{Tr}\Sigma_p \log\Sigma_q - \operatorname{Tr}\Sigma_p \log\Gamma$$

$$= \int_0^{+\infty}\left(\operatorname{Tr}\Sigma_p(\Gamma + \beta I)^{-1} - \operatorname{Tr}\Sigma_p(\Sigma_q + \beta I)^{-1}\right)d\beta$$

$$= \alpha\int_0^{+\infty}\operatorname{Tr}\Sigma_p(\Sigma_q + \beta I)^{-1}(\Sigma_q - \Sigma_p)(\Sigma_q + \beta I)^{-1}d\beta$$

$$- \alpha^2\int_0^{+\infty}\operatorname{Tr}\Sigma_p(\Sigma_q + \beta I)^{-1}(\Sigma_q - \Sigma_p)(\Gamma + \beta I)^{-1}(\Sigma_q - \Sigma_p)(\Sigma_q + \beta I)^{-1}d\beta$$

$$:= (a) - (b).$$

We will first upper bound $(a)$ in absolute value, since it is not necessarily positive. We first bound

$$(\Sigma_q + \beta I)^{-1}\Sigma_q(\Sigma_q + \beta I)^{-1} \preccurlyeq (\Sigma_q + \beta I)^{-1} \text{ and } (\Sigma_q + \beta I)^{-1}\Sigma_p(\Sigma_q + \beta I)^{-1} \preccurlyeq \frac{1}{\mu}(\Sigma_q + \beta I)^{-1},$$

where we used for the second term the matrix inequalities $\Sigma_p \preccurlyeq \frac{1}{\mu}\Sigma_q$. Hence, we have

$$|\operatorname{Tr}\Sigma_p(\Sigma_q + \beta I)^{-1}(\Sigma_q - \Sigma_p)(\Sigma_q + \beta I)^{-1}| = |\operatorname{Tr}\Sigma_p^{\frac{1}{2}}(\Sigma_q + \beta I)^{-1}(\Sigma_q - \Sigma_p)(\Sigma_q + \beta I)^{-1}\Sigma_p^{\frac{1}{2}}|$$

$$\leqslant \operatorname{Tr}\Sigma_p^{\frac{1}{2}}(\Sigma_q + \beta I)^{-1}\Sigma_q(\Sigma_q + \beta I)^{-1}\Sigma_p^{\frac{1}{2}} + \operatorname{Tr}\Sigma_p^{\frac{1}{2}}(\Sigma_q + \beta I)^{-1}\Sigma_p(\Sigma_q + \beta I)^{-1}\Sigma_p^{\frac{1}{2}}$$

$$\leqslant \operatorname{Tr}\Sigma_p^{\frac{1}{2}}(\Sigma_q + \beta I)^{-1}\Sigma_p^{\frac{1}{2}} + \frac{1}{\mu}\operatorname{Tr}\Sigma_p^{\frac{1}{2}}(\Sigma_q + \beta I)^{-1}\Sigma_p^{\frac{1}{2}}$$

$$= \left(1 + \frac{1}{\mu}\right)\operatorname{Tr}\Sigma_p(\Sigma_q + \beta I)^{-1}.$$

We can then upper bound $|(a)|$ as:

$$\left|\alpha\int_0^{+\infty}\operatorname{Tr}\Sigma_p(\Sigma_q + \beta I)^{-1}(\Sigma_q - \Sigma_p)(\Sigma_q + \beta I)^{-1}d\beta\right| \leqslant \alpha\left(1 + \frac{1}{\mu}\right)|\operatorname{Tr}\Sigma_p \log\Sigma_q|.$$

We now turn to $(b)$ which we can upper bound without absolute value since it is a positive term. Since $\Gamma \succcurlyeq (1-\alpha)\Sigma_q$, $\alpha \in [0, 1]$ and we are dealing with p.s.d. operators, we can bound the inverse as $(\Gamma + \beta I)^{-1} \preccurlyeq \frac{1}{1-\alpha}(\Sigma_q + \frac{\beta}{1-\alpha}I)^{-1} \preccurlyeq \frac{1}{1-\alpha}(\Sigma_q + \beta I)^{-1}$ and so, using $\operatorname{Tr}(AM) \leq \operatorname{Tr}(BM)$ for $A \preccurlyeq B$ and $M \succcurlyeq 0$, we have:

$$\operatorname{Tr}\Sigma_p(\Sigma_q + \beta I)^{-1}(\Sigma_q - \Sigma_p)(\Gamma + \beta I)^{-1}(\Sigma_q - \Sigma_p)(\Sigma_q + \beta I)^{-1}$$

$$\leqslant \frac{1}{1-\alpha}\operatorname{Tr}\Sigma_p(\Sigma_q + \beta I)^{-1}(\Sigma_q - \Sigma_p)(\Sigma_q + \beta I)^{-1}(\Sigma_q - \Sigma_p)(\Sigma_q + \beta I)^{-1}.$$

We will split the r.h.s. of the previous inequality in four terms, involving twice $\Sigma_q$, two cross terms (bounded similarly) with $\Sigma_p, \Sigma_q$ and twice $\Sigma_p$. We have for the first one:

$$\operatorname{Tr} \Sigma_p^{\frac{1}{2}}(\Sigma_q + \beta I)^{-1}\Sigma_q(\Sigma_q + \beta I)^{-1}\Sigma_q(\Sigma_q + \beta I)^{-1}\Sigma_p^{\frac{1}{2}} \leqslant \operatorname{Tr} \Sigma_p^{\frac{1}{2}}(\Sigma_q + \beta I)^{-1}\Sigma_q(\Sigma_q + \beta I)^{-1}\Sigma_p^{\frac{1}{2}}$$
$$\leqslant \operatorname{Tr} \Sigma_p(\Sigma_q + \beta I)^{-1}.$$

For the cross-term we have:

$$- \operatorname{Tr} \Sigma_p(\Sigma_q + \beta I)^{-1}\Sigma_p(\Sigma_q + \beta I)^{-1}\Sigma_q(\Sigma_q + \beta I)^{-1} \leqslant 0.$$

and for the last term we have:

$$\operatorname{Tr} \Sigma_p(\Sigma_q + \beta I)^{-1}\Sigma_p(\Sigma_q + \beta I)^{-1}\Sigma_p(\Sigma_q + \beta I)^{-1} \leqslant \frac{1}{\mu^2} \operatorname{Tr} \Sigma_p(\Sigma_q + \beta I)^{-1}.$$

Finally, combining our bounds for the terms in $(b)$ and integrating with respect to $\beta$, we get

$$\alpha^2 \int_0^{+\infty} \operatorname{Tr} \Sigma_p(\Sigma_q + \beta I)^{-1}(\Sigma_q - \Sigma_p)(\Gamma + \beta I)^{-1}(\Sigma_q - \Sigma_p)(\Sigma_q + \beta I)^{-1}d\beta$$
$$\leqslant \frac{\alpha^2}{1-\alpha}\left(1 + \frac{1}{\mu^2}\right) \operatorname{Tr} \Sigma_p \log \Sigma_q.$$

Adding the bounds on (a) and (b) concludes the proof.

### B.3 Proof of Proposition 4

#### B.3.1 Intermediate result 1: Concentration of sums of random self-adjoint operators

**Lemma 10.** *Assume the conditions of Proposition 4 hold. Denote $\hat{\Gamma} = \alpha\Sigma_{\hat{p}} + (1-\alpha)\Sigma_{\hat{q}}$ and $\Gamma$ its population counterpart. Let $\beta > 0$, $D = (\Gamma + \beta I)^{-\frac{1}{2}}(\hat{\Gamma} - \Gamma)(\Gamma + \beta I)^{-\frac{1}{2}}$ and $C(\beta) = \sup_{x \in \mathbb{R}^d} \langle \varphi(x), (\Sigma_p + \beta I)^{-1}\varphi(x)\rangle_{\mathcal{H}}$. Then, for $0 < u < 1$,*

$$\mathbb{P}(\lambda_{\max}(D) > u) \leqslant \frac{2}{\mu}C(\beta)\left(1 + \frac{48}{u^4(m \wedge n)}\left(\frac{C(\beta)}{\mu} + \frac{u}{6}\right)^2\right)\exp\left(-\frac{(m \wedge n)u^2}{8(\frac{C(\beta)}{\mu} + \frac{u}{6})}\right)$$

*where $\lambda_{\max}(D)$ is the maximal eigenvalue of $(D^2)^{1/2}$.*

*Proof.* Denoting

$$X_i = (\Gamma + \beta I)^{-1/2}\varphi(x_i)\varphi(x_i)^*(\Gamma + \beta I)^{-1/2} \text{ and } Y_j = (\Gamma + \beta I)^{-1/2}\varphi(y_j)\varphi(y_j)^*(\Gamma + \beta I)^{-1/2},$$

we have $(\Gamma + \beta I)^{-1/2}\Sigma_p(\Gamma + \beta I)^{-1/2} = \mathbb{E}[X]$, $(\Gamma + \beta I)^{-1/2}\Sigma_q(\Gamma + \beta I)^{-1/2} = \mathbb{E}[Y]$ and so $(\Gamma + \beta I)^{-1/2}\Gamma(\Gamma + \beta I)^{-1/2} = \alpha\mathbb{E}[X] + (1-\alpha)\mathbb{E}[Y]$. We can thus write

$$D = \frac{\alpha}{n}\sum_{i=1}^n (X_i - \mathbb{E}[X]) + \frac{1-\alpha}{m}\sum_{j=1}^m (Y_j - \mathbb{E}[Y]).$$

However, for two operators $A$ and $B$ we have $\|A + B\|_{op} \leqslant \|A\|_{op} + \|B\|_{op}$ which means that $\lambda_{\max}(A + B) \leqslant \lambda_{\max}(A) + \lambda_{\max}(B)$. Then,

$$\lambda_{\max}(D) \leqslant \lambda_{\max}\left(\frac{1}{n}\sum_{i=1}^n \alpha X_i - \mathbb{E}[\alpha X]\right) + \lambda_{\max}\left(\frac{1}{m}\sum_{j=1}^m (1-\alpha)Y_j - \mathbb{E}[(1-\alpha)Y]\right),$$

which is equivalent to

$$\lambda_{\max}(D) \leqslant \lambda_{\max}\left(\alpha(\Gamma + \beta I)^{-\frac{1}{2}}(\Sigma_{\hat{p}} - \Sigma_p)(\Gamma + \beta I)^{-\frac{1}{2}}\right)$$
$$+ \lambda_{\max}\left((1-\alpha)(\Gamma + \beta I)^{-\frac{1}{2}}(\Sigma_{\hat{q}} - \Sigma_q)(\Gamma + \beta I)^{-\frac{1}{2}}\right).$$

The quantities $\lambda_{\max}(D_p) := \lambda_{\max}\left(\alpha(\Gamma + \beta I)^{-\frac{1}{2}}(\Sigma_{\hat{p}} - \Sigma_p)(\Gamma + \beta I)^{-\frac{1}{2}}\right)$ and $\lambda_{\max}(D_q) := \lambda_{\max}\left((1-\alpha)(\Gamma + \beta I)^{-\frac{1}{2}}(\Sigma_{\hat{q}} - \Sigma_q)(\Gamma + \beta I)^{-\frac{1}{2}}\right)$ are positive so

$$\mathbb{P}\left(\lambda_{\max}(D) > u\right) \leqslant \mathbb{P}\left(\lambda_{\max}(D_p) > \frac{u}{2}\right) + \mathbb{P}\left(\lambda_{\max}(D_q) > \frac{u}{2}\right).$$

Then Bach [2022, Lemma 2] can be applied twice to $\tilde{\varphi}(x) = \sqrt{\alpha}(\Gamma + \beta I)^{-1/2}\varphi(x)$ and to $\tilde{\tilde{\varphi}}(y) = \sqrt{(1-\alpha)}(\Gamma + \beta I)^{-1/2}\varphi(y)$.

For the term with $D_p$,

$$\|\tilde{\varphi}(x)\|_{\mathcal{H}}^2 = \|\sqrt{\alpha}(\Gamma + \beta I)^{-1/2}\varphi(x)\|_{\mathcal{H}}^2 = \alpha\langle\varphi(x), (\Gamma + \beta I)^{-1}\varphi(x)\rangle_{\mathcal{H}}$$
$$\leqslant \langle\varphi(x), (\Sigma_p + \beta I)^{-1}\varphi(x)\rangle_{\mathcal{H}} \leqslant C(\beta),$$

where we used $\Gamma \succcurlyeq \alpha\Sigma_p$ and define $C(\beta) = \sup_{x\in\mathbb{R}^d}\langle\varphi(x), (\Sigma_p + \beta I)^{-1}\varphi(x)\rangle_{\mathcal{H}}$. Using the same inequality, we also obtain

$$\mathrm{Tr}(\Gamma + \beta I)^{-1/2}\alpha\Sigma_p(\Gamma + \beta I)^{-1/2} \leqslant \mathrm{Tr}(\Sigma_p + \beta I)^{-1/2}\Sigma_p(\Sigma_p + \beta I)^{-1/2}$$
$$= \mathrm{Tr}\,\Sigma_p(\Sigma_p + \beta I)^{-1}$$
$$= \int\langle\varphi(x)(\Sigma_p + \beta I)^{-1}\varphi(x)\rangle_{\mathcal{H}}dp(x) \leqslant C(\beta),$$

and

$$\lambda_{max}((\Gamma + \beta I)^{-1/2}\alpha\Sigma_p(\Gamma + \beta I)^{-1/2}) = \|(\Gamma + \beta I)^{-1/2}\alpha\Sigma_p(\Gamma + \beta I)^{-1/2}\|_{op} \leqslant 1.$$

Then,

$$\mathbb{P}\left(\lambda_{\max}(D_p) > \frac{u}{2}\right) \leqslant C(\beta)\left(1 + \frac{48}{u^4 n^2}\left(C(\beta) + \frac{u}{6}\right)^2\right)\exp\left(-\frac{nu^2}{8(C(\beta) + \frac{u}{6})}\right).$$

For the term with $D_q$, using the matrix inequalities $\Gamma \succcurlyeq (1-\alpha)\Sigma_q$ and $\Sigma_q \succcurlyeq \mu\Sigma_p$, with similar computations we get

$$\|\tilde{\tilde{\varphi}}(y)\|_{\mathcal{H}}^2 \leqslant \frac{1}{\mu}C(\beta), \quad \mathrm{Tr}(\Gamma + \beta I)^{-1/2}(1-\alpha)\Sigma_q(\Gamma + \beta I)^{-1/2} \leqslant \frac{1}{\mu}C(\beta),$$

$$\text{and } \lambda_{\max}\left((\Gamma + \beta I)^{-1/2}(1-\alpha)\Sigma_q(\Gamma + \beta I)^{-1/2}\right) \leqslant 1.$$

Then,

$$\mathbb{P}\left(\lambda_{\max}(D_q) > \frac{u}{2}\right) \leqslant \frac{C(\beta)}{\mu}\left(1 + \frac{48}{u^4 m}\left(\frac{C(\beta)}{\mu} + \frac{u}{6}\right)^2\right)\exp\left(-\frac{mu^2}{8(\frac{C(\beta)}{\mu} + \frac{u}{6})}\right).$$

We can combine both results on $D_p$ and $D_q$ and use $\mu \leq 1$ to get

$$\mathbb{P}\left(\lambda_{\max}(D) > u\right) \leqslant \frac{2}{\mu}C(\beta)\left(1 + \frac{48}{u^4(m\wedge n)^2}\left(\frac{C(\beta)}{\mu} + \frac{u}{6}\right)^2\right)\exp\left(-\frac{(m\wedge n)u^2}{8(\frac{C(\beta)}{\mu} + \frac{u}{6})}\right). \qquad \square$$

### B.3.2 Intermediate result 2: Degrees of freedom estimation

The Proposition below adapts the proof of Bach [2022, Proposition 15] to bound the cross terms between the empirical covariance operators of $p$ and $\alpha p + (1-\alpha)q$.

**Proposition 11** (Estimation of skewed degrees of freedom). *Assume the conditions of Proposition 4 hold. Denote $\hat{\Gamma} = \alpha\Sigma_{\hat{p}} + (1-\alpha)\Sigma_{\hat{q}}$ and $\Gamma$ its population counterpart. We have:*

$$\left|\mathbb{E}\left[\mathrm{Tr}\,\Sigma_p(\Gamma + \beta I)^{-1} - \mathrm{Tr}\,\Sigma_{\hat{p}}(\hat{\Gamma} + \beta I)^{-1}\right]\right|$$
$$\leqslant \left(\frac{12}{\alpha\mu}n\exp\left(-\frac{m\wedge n}{16C(\beta)}\right) + \frac{6}{\mu}\sqrt{\left(\frac{1}{n} + \frac{1}{m}\right)}\right)C(\beta) + \frac{28}{\alpha\mu^2}\left(\frac{1}{n} + \frac{1}{m}\right)C(\beta)^2. \quad (15)$$

*where $C(\beta) = \sup_{x\in\mathbb{R}^d}\langle\varphi(x), (\Sigma_p + \beta I)^{-1}\varphi(x)\rangle_{\mathcal{H}}$ is supposed to be inferior to $\frac{\mu(m\wedge n)}{24}$.*

*Proof.* We will denote

$$A := \operatorname{Tr} \Sigma_p (\Gamma + \beta I)^{-1} - \operatorname{Tr} \Sigma_{\hat{p}} (\hat{\Gamma} + \beta I)^{-1}$$
$$= \operatorname{Tr}(\Sigma_p - \Sigma_{\hat{p}})(\Gamma + \beta I)^{-1} + \operatorname{Tr} \Sigma_{\hat{p}} (\Gamma + \beta I)^{-1}(\Gamma - \hat{\Gamma})(\hat{\Gamma} + \beta I)^{-1}. \tag{16}$$

In expectation, the first term in (16) can be upper-bounded as follows, using that $\operatorname{Tr}\big((\Sigma_p - \Sigma_{\hat{p}})(\Gamma + \beta I)^{-1}\big)$ is the sum of zero-mean random variables:

$$
\begin{aligned}
|\mathbb{E} \operatorname{Tr}(\Sigma_p - \Sigma_{\hat{p}})(\Gamma + \beta I)^{-1}| &\leqslant \sqrt{\mathbb{E}[(\operatorname{Tr}(\Sigma_p - \Sigma_{\hat{p}})(\Gamma + \beta I)^{-1})^2]} \\
&= \sqrt{\frac{1}{n} \operatorname{Var}[\langle \varphi(x), (\Gamma + \beta I)^{-1}\varphi(x)\rangle_{\mathcal{H}}]} \\
&\leqslant \sqrt{\frac{1}{n}\mathbb{E}[\langle \varphi(x), (\Gamma + \beta I)^{-1}\varphi(x)\rangle_{\mathcal{H}}^2]} \\
&\leqslant \frac{1}{\mu(1-\alpha)}\sqrt{\frac{1}{n}\mathbb{E}[\langle \varphi(x), (\Sigma_p + \frac{\beta}{\mu(1-\alpha)}I)^{-1}\varphi(x)\rangle_{\mathcal{H}}^2]} \\
&\leqslant \sqrt{\frac{1}{n}}\frac{1}{\mu(1-\alpha)}C\left(\frac{\beta}{\mu(1-\alpha)}\right) \\
&\leqslant \sqrt{\frac{1}{n}\frac{2}{\mu}}C(\beta),
\end{aligned}
\tag{17}
$$

where the third inequality uses $\Gamma \succcurlyeq (1-\alpha)\Sigma_q \succcurlyeq \mu(1-\alpha)\Sigma_p$ and the fourth uses the definition of $C(\beta)$ for $\beta > 0$. The last inequality is due to the facts that $\alpha \leqslant \frac{1}{2}$ and so $\frac{1}{1-\alpha} \leqslant 2$, and also that $\mu \leqslant 1$ so $\frac{\beta}{\mu(1-\alpha)} \geqslant \beta$ and because $\beta \mapsto C$ is non increasing on $]0, +\infty[$, we have $C\left(\frac{\beta}{\mu(1-\alpha)}\right) \leqslant C(\beta)$. These simplifications are used many times in this proof.

The second term in (16) can be written as follows. Consider $D = (\Gamma + \beta I)^{-\frac{1}{2}}(\Gamma - \hat{\Gamma})(\Gamma + \beta I)^{-\frac{1}{2}}$ defined in Lemma 10 and consider the case where $\lambda_{max}(D) \leqslant u < 1$. Then $D \prec I$. Using the identity $D(I-D)^{-1} = D(I-D)^{-1}(D + I - D)$, as in the proof of [Rudi and Rosasco, 2017], we can write

$$
\begin{aligned}
B &:= \operatorname{Tr} \Sigma_{\hat{p}} (\Gamma + \beta I)^{-1}(\Gamma - \hat{\Gamma})(\hat{\Gamma} + \beta I)^{-1} \\
&= \operatorname{Tr} \Sigma_{\hat{p}} (\Gamma + \beta I)^{-\frac{1}{2}} D(I-D)^{-1}(\Gamma + \beta I)^{-\frac{1}{2}} \\
&= \operatorname{Tr} \Sigma_{\hat{p}} (\Gamma + \beta I)^{-\frac{1}{2}} D(I-D)^{-1}D(\Gamma + \beta I)^{-\frac{1}{2}} + \operatorname{Tr} \Sigma_{\hat{p}}(\Gamma + \beta I)^{-\frac{1}{2}}D(\Gamma + \beta I)^{-\frac{1}{2}}. \tag{18}
\end{aligned}
$$

We have for the first term in (18), using $\hat{\Gamma} \succcurlyeq \alpha \Sigma_{\hat{p}}$:

$$
\begin{aligned}
\operatorname{Tr} \Sigma_{\hat{p}}(\Gamma + \beta I)^{-\frac{1}{2}}D(\Gamma + \beta I)^{-\frac{1}{2}} &= \operatorname{Tr} D^{\frac{1}{2}}(\Gamma + \beta I)^{-\frac{1}{2}}\Sigma_{\hat{p}}(\Gamma + \beta I)^{-\frac{1}{2}}D^{\frac{1}{2}} \\
&\leqslant \frac{1}{\alpha}\operatorname{Tr} D^{\frac{1}{2}}(\Gamma + \beta I)^{-\frac{1}{2}}\hat{\Gamma}(\Gamma + \beta I)^{-\frac{1}{2}}D^{\frac{1}{2}},
\end{aligned}
$$

and, by the definition of $D$ above,

$$-\lambda_{\max}(D)I \preccurlyeq (\Gamma + \beta I)^{-\frac{1}{2}}(\Gamma - \hat{\Gamma})(\Gamma + \beta I)^{-\frac{1}{2}} \preccurlyeq \lambda_{\max}(D)I$$

where $\lambda_{\max}(D)$ is the absolute value of the maximal eigenvalue of $(D^2)^{\frac{1}{2}}$. So

$$(\Gamma + \beta I)^{-\frac{1}{2}}\hat{\Gamma}(\Gamma + \beta I)^{-\frac{1}{2}} \preccurlyeq (\lambda_{\max}(D) + 1)I. \tag{19}$$

In this case,

$$\frac{1}{\alpha}\operatorname{Tr} D^{\frac{1}{2}}(\Gamma + \beta I)^{-\frac{1}{2}}\hat{\Gamma}(\Gamma + \beta I)^{-\frac{1}{2}}D^{\frac{1}{2}} \leqslant \frac{1 + \lambda_{\max}(D)}{\alpha}\operatorname{Tr} D. \tag{20}$$

Still considering that $\lambda_{\max}(D) < 1$ we have for the second term in (18):

$$\operatorname{Tr}\Sigma_{\hat{p}}(\Gamma + \beta I)^{-\frac{1}{2}}D(I-D)^{-1}D(\Gamma + \beta I)^{-\frac{1}{2}}$$

$$\leqslant \|(\Gamma + \beta I)^{-\frac{1}{2}}\Sigma_{\hat{p}}(\Gamma + \beta I)^{-\frac{1}{2}}\|_{op}\operatorname{Tr}\left(D^2(I-D)^{-1}\right)$$

$$\leqslant \frac{1 + \lambda_{\max}(D)}{\alpha}\|(I-D)^{-1}\|_{op}\operatorname{Tr}D^2$$

$$= \frac{1 + \lambda_{\max}(D)}{\alpha(1 - \lambda_{\max}(D))}\operatorname{Tr}D^2, \tag{21}$$

where in the second inequality we used $\Sigma_{\hat{p}} \preccurlyeq \frac{1}{\alpha}\hat{\Gamma}$ and (19). Let $0 < u < 1$. Combining (17), (20) and (21), we have:

$$|A| = \mathbb{1}_{\lambda_{\max}(D)>u}|A| + \mathbb{1}_{\lambda_{\max}(D)\leqslant u}|A|$$

$$\leqslant \mathbb{1}_{\lambda_{\max}(D)>u}|A| + \mathbb{1}_{\lambda_{\max}(D)\leqslant u}\left(\frac{1+u}{\alpha(1-u)}\operatorname{Tr}D^2 + \frac{1+u}{\alpha}\operatorname{Tr}D + \sqrt{\frac{1}{n}\frac{2}{\mu}}C(\beta)\right)$$

$$\leqslant \mathbb{1}_{\lambda_{\max}(D)>u}|A| + \frac{1+u}{\alpha(1-u)}\operatorname{Tr}D^2 + \frac{1+u}{\alpha}\operatorname{Tr}D + \sqrt{\frac{1}{n}\frac{2}{\mu}}C(\beta). \tag{22}$$

We now bound the first term of (22) by upperbounding, going back to the formula given in (16). Using $\Gamma \succcurlyeq \mu(1-\alpha)\Sigma_p$, the term $\operatorname{Tr}\Sigma_p(\Gamma+\beta I)^{-1}$ is bounded by $\frac{1}{\mu(1-\alpha)}C\left(\frac{\beta}{\mu(1-\alpha)}\right) \leqslant \frac{2}{\mu}C(\beta) \leqslant \frac{n\wedge m}{12}$ by hypothesis. And with $\hat{\Gamma} \succcurlyeq \frac{1}{\alpha}\Sigma_{\hat{p}}$, we have both $\operatorname{Tr}\Sigma_{\hat{p}}(\hat{\Gamma}+\beta I)^{-1} \leqslant \frac{1}{\alpha}\operatorname{Tr}\Sigma_{\hat{p}}(\Sigma_{\hat{p}}+\frac{\beta}{\alpha}I)^{-1} \leqslant \frac{n}{\alpha}$ and $\operatorname{Tr}\Sigma_p(\Gamma+\beta I)^{-1} \leqslant \frac{1}{(1-\alpha)\mu}C(\beta) \leqslant \frac{2}{\mu}C(\beta) \leqslant \frac{n\wedge m}{12}$. Then, almost surely, $|A| \leqslant \max\left\{\frac{n}{\alpha}, \frac{n\wedge m}{12}\right\} \leqslant \frac{2n}{\alpha}$. Then, (22) becomes:

$$\mathbb{E}|A| \leqslant \mathbb{P}(\lambda_{\max}(D) > u)\frac{2n}{\alpha} + \mathbb{E}\left[\frac{1+u}{\alpha(1-u)}\operatorname{Tr}D^2 + \frac{1+u}{\alpha}\operatorname{Tr}D\right] + \sqrt{\frac{1}{n}\frac{2}{\mu}}C(\beta).$$

With Lemma 10 we have

$$\mathbb{P}(\lambda_{\max}(D) > u) \leqslant \frac{2}{\mu}C(\beta)\left(1 + \frac{48}{u^4(m\wedge n)^2}\left(\frac{C(\beta)}{\mu} + \frac{u}{6}\right)^2\right)\exp\left(-\frac{(m\wedge n)u^2}{8(\frac{C(\beta)}{\mu} + \frac{u}{6})}\right).$$

Using the hypothesis that $C(\beta) \leqslant \frac{\mu(n\wedge m)}{24}$,

$$\mathbb{E}|A| \leqslant \frac{4n}{\mu\alpha}C(\beta)\left(1 + \frac{48}{u^4(m\wedge n)^2}\left(\frac{m\wedge n}{24} + \frac{u}{6}\right)^2\right)\exp\left(-\frac{(m\wedge n)u^2}{8(\frac{C(\beta)}{\mu} + \frac{u}{6})}\right)$$

$$+ \mathbb{E}\left[\frac{1+u}{\alpha(1-u)}\operatorname{Tr}D^2 + \frac{1+u}{\alpha}\operatorname{Tr}D\right] + \sqrt{\frac{1}{n}\frac{2}{\mu}}C(\beta). \tag{23}$$

We now turn to bounding $\mathbb{E}[\operatorname{Tr}D]$. We have

$$\mathbb{E}[\operatorname{Tr}D] \leqslant \sqrt{\mathbb{E}[(\operatorname{Tr}(\Gamma+\beta I)^{-\frac{1}{2}}(\Gamma - \hat{\Gamma})(\Gamma+\beta I)^{-\frac{1}{2}})^2]} \tag{24}$$

and denoting

$$X_i = \operatorname{Tr}(\Gamma+\beta I)^{-\frac{1}{2}}(\Gamma - \varphi(x_i)\varphi(x_i)^*)(\Gamma+\beta I)^{-\frac{1}{2}},$$

$$Y_j = \operatorname{Tr}(\Gamma+\beta I)^{-\frac{1}{2}}(\Gamma - \varphi(y_j)\varphi(y_j)^*)(\Gamma+\beta I)^{-\frac{1}{2}},$$

we have

$$\mathbb{E}[(\operatorname{Tr}(\Gamma+\beta I)^{-\frac{1}{2}}(\Gamma - \hat{\Gamma})(\Gamma+\beta I)^{-\frac{1}{2}})^2] = \frac{\alpha^2}{n^2}\sum_{i,k=1}^{n}\mathbb{E}[(\mathbb{E}[X] - X_i) \times (\mathbb{E}[X] - X_k)]$$

$$+ \frac{(1-\alpha)^2}{m^2}\sum_{j,l=1}^{m}\mathbb{E}[\mathbb{E}[Y] - Y_j)(\mathbb{E}[Y] - Y_l)] + \frac{\alpha(1-\alpha)}{nm}\sum_{i,j}\mathbb{E}[(\mathbb{E}[X] - X_i)(\mathbb{E}[Y] - Y_j)].$$

The variables $X_1, ..., X_n, Y_1, ..., Y_m$ are independent so we get

$$\mathbb{E}[(\text{Tr}(\Gamma + \beta I)^{-\frac{1}{2}}(\Gamma - \hat{\Gamma})(\Gamma + \beta I)^{-\frac{1}{2}})^2]$$

$$= \frac{\alpha^2}{n^2} \sum_{i,k=1}^{n} \mathbb{E}[X_i X_k] + \frac{(1-\alpha)^2}{m^2} \sum_{j,l=1}^{n} \mathbb{E}[Y_j Y_k] + \frac{\alpha(1-\alpha)}{nm} \sum_{i,j=1}^{n} \mathbb{E}[X_i Y_j]$$

$$= \frac{\alpha^2}{n} \mathbb{E}[X^2] + \frac{\alpha^2(n-1)}{n} \mathbb{E}[X]^2 + \frac{(1-\alpha)^2}{m} \mathbb{E}[Y^2] + \frac{(1-\alpha)^2(m-1)}{m} \mathbb{E}[Y]^2$$
$$\quad + 2\alpha(1-\alpha)\mathbb{E}[X]\mathbb{E}[Y]$$

$$= \frac{\alpha^2}{n}(\mathbb{E}[X^2] - \mathbb{E}[X]^2) + \frac{(1-\alpha)^2}{m}(\mathbb{E}[Y^2] - \mathbb{E}[Y]^2) + (\alpha\mathbb{E}[X] + (1-\alpha)\mathbb{E}[Y])^2$$

$$= \frac{\alpha^2}{n}\text{Var}[X] + \frac{(1-\alpha)^2}{m}\text{Var}[Y].$$

The last equality is due to $\mathbb{E}[\alpha\varphi(x)\varphi(x)^* + (1-\alpha)\varphi(y)\varphi(y)^*] = 0$. We have

$$\text{Var}[X] = \text{Var}[\text{Tr}(\Gamma + \beta I)^{-\frac{1}{2}}(\Gamma - \varphi(x)\varphi(x)^*)(\Gamma + \beta I)^{-\frac{1}{2}}]$$

$$= \text{Var}[\text{Tr}(\Gamma + \beta I)^{-\frac{1}{2}}\varphi(x)\varphi(x)^*(\Gamma + \beta I)^{-\frac{1}{2}}]$$

$$\leqslant \mathbb{E}[(\text{Tr}(\Gamma + \beta I)^{-\frac{1}{2}}\varphi(x)\varphi(x)^*(\Gamma + \beta I)^{-\frac{1}{2}})^2]$$

$$= \mathbb{E}[(\langle\varphi(x), (\Gamma + \beta I)^{-1}\varphi(x)\rangle_{\mathcal{H}}^2]$$

and the equivalent inequality is also verified for $Y$. Then,

$$\mathbb{E}[(\text{Tr}(\Gamma + \beta I)^{-\frac{1}{2}}(\Gamma - \hat{\Gamma})(\Gamma + \beta I)^{-\frac{1}{2}})^2]$$

$$\leqslant \frac{\alpha^2}{n}\mathbb{E}[\langle\varphi(x), (\Gamma + \beta I)^{-1}\varphi(x)\rangle_{\mathcal{H}}^2] + \frac{(1-\alpha)^2}{m}\mathbb{E}[\langle\varphi(y), (\Gamma + \beta I)^{-1}\varphi(y)\rangle_{\mathcal{H}}^2]$$

$$\leqslant \frac{1}{\mu^2(1-\alpha)^2}(\frac{\alpha^2}{n} + \frac{(1-\alpha)^2}{m})C\left(\frac{\beta}{\mu(1-\alpha)}\right)^2 \leqslant \frac{4}{\mu^2}\left(\frac{1}{n} + \frac{1}{m}\right)C(\beta)^2,$$

so we (24) becomes

$$\mathbb{E}[\text{Tr}\, D] \leqslant \frac{2}{\mu}\sqrt{\left(\frac{1}{n} + \frac{1}{m}\right)}C(\beta).$$

Using similar calculations, we obtain

$$\mathbb{E}[\text{Tr}\, D^2] \leqslant \frac{4}{\mu^2}\left(\frac{1}{n} + \frac{1}{m}\right)C(\beta)^2.$$

Finally, (23) becomes

$$\mathbb{E}|A| \leqslant \frac{4}{\alpha\mu}n\left(1 + \frac{48}{u^4(m \wedge n)^2}\left(\frac{m \wedge n}{24} + \frac{u}{6}\right)^2\right)\exp\left(-\frac{(m \wedge n)u^2}{8(\frac{C(\beta)}{\mu} + \frac{u}{6})}\right)C(\beta)$$

$$+ \frac{1+u}{\alpha(1-u)}\frac{4}{\mu^2}\left(\frac{1}{n} + \frac{1}{m}\right)C(\beta)^2 + \left(\frac{1+u}{\alpha}\sqrt{\left(\frac{1}{n} + \frac{1}{m}\right)} + \sqrt{\frac{1}{n}}\right)\frac{2}{\mu}C(\beta).$$

Taking $u = \frac{3}{4}$ we get the final bound

$$\mathbb{E}|A| \leqslant \frac{4}{\alpha\mu}n\left(1 + \frac{160}{(m \wedge n)^2}\left(\frac{m \wedge n}{24} + \frac{1}{8}\right)^2\right)\exp\left(-\frac{9(m \wedge n)}{16(8\frac{C(\beta)}{\mu} + 1)}\right)C(\beta)$$

$$+ \frac{28}{\alpha\mu^2}\left(\frac{1}{n} + \frac{1}{m}\right)C(\beta)^2 + \frac{11}{2\mu}\sqrt{\left(\frac{1}{n} + \frac{1}{m}\right)}C(\beta)$$

$$\leqslant \left(\frac{12}{\alpha\mu}n\exp\left(-\mu\frac{m \wedge n}{16C(\beta)}\right) + \frac{6}{\mu}\sqrt{\left(\frac{1}{n} + \frac{1}{m}\right)}\right)C(\beta) + \frac{28}{\alpha\mu^2}\left(\frac{1}{n} + \frac{1}{m}\right)C(\beta)^2. \quad \square$$

### B.3.3  Final proof of Proposition 4

From [Bach, 2022, Proposition 7] we already have a bound on the entropy term:

$$\mathbb{E}[|\operatorname{Tr}(\Sigma_{\hat{p}}\log\Sigma_{\hat{p}}) - \operatorname{Tr}(\Sigma_p\log\Sigma_p)|] \leqslant \frac{1 + c(8\log n)^2}{n} + \frac{17}{\sqrt{n}}(2\sqrt{c} + \log n). \tag{25}$$

In the following we will closely follow the proof of [Bach, 2022, Proposition 7] in order to bound the cross terms difference. We can write with the integral representation in Bach [2022] (Eq 5),

$$\operatorname{Tr}\Sigma_{\hat{p}}\log\hat{\Gamma} - \operatorname{Tr}\Sigma_p\log\Gamma = \int_0^{+\infty} \operatorname{Tr}\Sigma_p(\Gamma + \beta I)^{-1} - \operatorname{Tr}\Sigma_{\hat{p}}(\hat{\Gamma} + \beta I)^{-1}d\beta.$$

We will treat separately the integral part close to infinity, the one close to zero and the central part. Let $\beta_1 > \beta_0 > 0$. From $\beta_1$ to infinity we have

$$\int_{\beta_1}^{+\infty} \operatorname{Tr}\Sigma_p(\Gamma + \beta I)^{-1} - \operatorname{Tr}\Sigma_{\hat{p}}(\hat{\Gamma} + \beta I)^{-1}d\beta = \operatorname{Tr}\Sigma_{\hat{p}}\log\left(\hat{\Gamma} + \beta_1 I\right) - \Sigma_p\log(\Gamma + \beta_1 I)$$

$$\leqslant \log(1 + \beta_1) - \log\beta_1 \leqslant \frac{1}{\beta_1}.$$

From $0$ to $\beta_0$ we have

$$\int_0^{\beta_0} \operatorname{Tr}\Sigma_p(\Gamma + \beta I)^{-1}d\beta \leqslant \int_0^{\beta_0} \operatorname{Tr}\Sigma_p((1 - \alpha)\mu\Sigma_p + \beta I)^{-1}d\beta$$

$$= \frac{1}{\mu(1 - \alpha)}\int_0^{\beta_0} \operatorname{Tr}\Sigma_p(\Sigma_p + \frac{\beta}{\mu(1 - \alpha)}I)^{-1}d\beta$$

$$\leqslant \frac{1}{\mu(1 - \alpha)}\int_0^{\beta_0} \sup_{x\in\mathbb{R}^d}\langle\varphi(x), (\Sigma_p + \frac{\beta}{\mu(1 - \alpha)}I)^{-1}\varphi(x)\rangle_{\mathcal{H}}d\beta \leqslant \frac{2}{\mu}\int_0^{\beta_0} C(\beta)d\beta,$$

where $C(\beta) = \sup_{x\in\mathbb{R}^d}\langle\varphi(x), (\Sigma_p + \beta I)^{-1}\varphi(x)\rangle_{\mathcal{H}}$. We also have

$$\int_0^{\beta_0} \operatorname{Tr}\Sigma_{\hat{p}}(\hat{\Gamma} + \beta I)^{-1}d\beta \leqslant \int_0^{\beta_0} \operatorname{Tr}\Sigma_{\hat{p}}(\alpha\Sigma_{\hat{p}} + \beta I)^{-1}d\beta \leqslant \frac{1}{\alpha}n\beta_0.$$

By Proposition 11 we have

$$\mathbb{E}|\operatorname{Tr}\Sigma_{\hat{p}}\log\hat{\Gamma} - \operatorname{Tr}\Sigma_p\log\Gamma|$$

$$\leqslant \frac{1}{\mu\beta_1} + \frac{1}{\alpha}n\mu\beta_0 + \int_0^{\beta_0} C(\beta)d\beta + \int_{\beta_0}^{\beta_1} \mathbb{E}|\operatorname{Tr}\Sigma_{\hat{p}}\log\hat{\Gamma} - \operatorname{Tr}\Sigma_p\log\Gamma|d\beta$$

$$\leqslant \frac{1}{\mu\beta_1} + \frac{1}{\alpha}n\mu\beta_0 + \int_0^{\beta_0} C(\beta)d\beta + \left(\frac{12}{\alpha\mu}n\exp\left(-\mu\frac{m\wedge n}{16C(\beta_0)}\right) + \frac{6}{\mu}\sqrt{\left(\frac{1}{n} + \frac{1}{m}\right)}\right)\int_{\beta_0}^{\beta_1} C(\beta)d\beta$$

$$+ \frac{28}{\alpha\mu^2}\left(\frac{1}{n} + \frac{1}{m}\right)\int_{\beta_0}^{\beta_1} C(\beta)^2 d\beta.$$

We now take $\beta_0$ such that $C(\beta_0) = \mu\frac{n\wedge m}{24\log(n)}$. The function $\beta\mapsto C(\beta)$ being non-increasing on $]0,\infty[$, the condition of Proposition 11, which is $C(\beta) \leqslant \frac{\mu(m\wedge n)}{24}$, is well satisfied between $\beta_0$ and $\beta_1$ for this choice of $\beta_0$. We then have

$$\frac{12}{\alpha\mu}n\exp\left(-\mu\frac{m\wedge n}{16C(\beta_0)}\right) \leqslant \frac{12}{\alpha\mu}n\exp\left(-\frac{24}{16}\log(n)\right) \leqslant \frac{12}{\alpha\mu}\frac{1}{\sqrt{n}} \leqslant \frac{12}{\alpha\mu}\sqrt{\frac{1}{n} + \frac{1}{m}},$$

and also, $\int_{\beta_0}^{\beta_1} C(\beta)^2 d\beta \leqslant c$. Then,

$$\mathbb{E}|\operatorname{Tr}\Sigma_{\hat{p}}\log\hat{\Gamma} - \operatorname{Tr}\Sigma_p\log\Gamma| \leqslant \frac{1}{\mu\beta_1} + \frac{1}{\alpha}n\mu\beta_0 + \int_0^{\beta_0} C(\beta)d\beta$$

$$+ \frac{18}{\alpha\mu}\sqrt{\left(\frac{1}{n} + \frac{1}{m}\right)}\int_{\beta_0}^{\beta_1} C(\beta)d\beta + \frac{28c}{\alpha\mu^2}\left(\frac{1}{n} + \frac{1}{m}\right).$$

The function $\beta \mapsto C(\beta)$ is decreasing so, $C(\beta)^2 \frac{\beta}{2} \leqslant \int_{\beta/2}^{\beta} C(\beta')^2 d\beta' \leqslant c$ and so, $C(\beta) \leqslant \sqrt{\frac{2c}{\beta}}$. We also deduce from that

$$\beta_0 \leqslant 2c \left( \frac{24 \log(n)}{\mu(m \wedge n)} \right)^2. \tag{26}$$

Hence

$$\frac{1}{\alpha} n \mu \beta_0 + \int_0^{\beta_0} C(\beta) d\beta \leqslant \frac{2c}{\alpha} n \left( \frac{24 \log(n)}{\mu(m \wedge n)} \right)^2 + \frac{96c \log(n)}{\mu(n \wedge m)}.$$

We now take $\beta_1 = \beta_0 + n$. We have

$$\int_{\beta_0}^{\beta_1} C(\beta) d\beta \leqslant \int_0^{\beta_0 + 1} C(\beta) d\beta + \log \frac{\beta_0 + n}{\beta_0 + 1} \leqslant \log n + 2 \sqrt{c(1 + \beta_0)}.$$

Then, plugging the bound on $\beta_0$ given by (26),

$$\mathbb{E} |\operatorname{Tr} \Sigma_{\hat{p}} \log \hat{\Gamma} - \operatorname{Tr} \Sigma_p \log \Gamma| \leqslant \frac{1}{\mu n} + \frac{2c}{\alpha} \frac{\times n (24 \log n)^2}{\mu(m \wedge n)^2} + \frac{96c \log(n)}{\mu(n \wedge m)}$$
$$+ \frac{18}{\alpha \mu} \sqrt{\left( \frac{1}{n} + \frac{1}{m} \right)} \left( \log n + 2 \sqrt{c(1 + \frac{2c}{\mu^2} \frac{(24 \log n)^2}{(m \wedge n)^2})} \right) + \frac{28c}{\alpha \mu^2} \left( \frac{1}{n} + \frac{1}{m} \right). \tag{27}$$

Concatenating with (25), we get

$$\mathbb{E} |\operatorname{KKL}_\alpha(\hat{p}||\hat{q}) - \operatorname{KKL}_\alpha(p||q)| \leqslant \frac{1 + \frac{1}{\mu} + c(8 \log n)^2}{n} + \frac{2c}{\mu \alpha} \frac{n(24 \log n)^2}{(m \wedge n)^2} + \frac{17}{\sqrt{n}} (2\sqrt{c} + \log n)$$
$$+ \frac{28c}{\alpha \mu^2} \left( \frac{1}{n} + \frac{1}{m} \right) + \frac{96c \log(n)}{\mu(n \wedge m)} + \frac{18}{\alpha \mu} \sqrt{\left( \frac{1}{n} + \frac{1}{m} \right)} \left( \log n + 2 \sqrt{c(1 + \frac{2c}{\mu^2} \frac{(24 \log n)^2}{(m \wedge n)^2})} \right). \tag{28}$$

Using increments such that $1 \leqslant \log n \leqslant (\log n)^2$, $\frac{1}{n} \leqslant \frac{1}{n \wedge m} \leqslant \frac{1}{\sqrt{m \wedge n}} \leqslant 1$ and $\alpha, \mu \leqslant 1$, we can upperbound (28) by a simpler bound, while still retaining the main convergence rates in $\frac{\log n}{\sqrt{m \wedge n}}$ and in $\frac{(\log n)^2}{m \wedge n}$. This bound is

$$\mathbb{E} |\operatorname{KKL}_\alpha(\hat{p}||\hat{q}) - \operatorname{KKL}_\alpha(p||q)| \leqslant \frac{35}{\sqrt{m \wedge n}} \frac{1}{\alpha \mu} (2\sqrt{c} + \log n)$$
$$+ \frac{1}{m \wedge n} \left( 1 + \frac{1}{\mu} + (24 \log n)^2 \frac{c}{\alpha \mu^2} (1 + \frac{n}{m \wedge m}) \right). \tag{29}$$

As mentioned in Remark 5, it is possible to re-write this proof without considering the assumptions that $p \ll q$ and $\frac{dp}{dq} \leq \frac{1}{\mu}$ and to derive a bound similar to this one but which scales in $O(\frac{1}{\alpha^2})$ instead of $O(\frac{1}{\alpha})$. This bound is

$$\mathbb{E} |\operatorname{KKL}_\alpha(\hat{p}||\hat{q}) - \operatorname{KKL}_\alpha(p||q)| \leqslant \frac{32}{\alpha \sqrt{m \wedge n}} (2\sqrt{c} + \log n)$$
$$+ \frac{2}{m \wedge n} \left( \frac{1}{\alpha} + \frac{c(26 \log n)^2}{\alpha^2} (1 + \frac{n}{m \wedge m}) \right). \tag{30}$$

To get this new upper bound, the operator inequality $\Gamma \succcurlyeq (1 - \alpha) \Sigma_q \succcurlyeq (1 - \alpha) \mu \Sigma_p$ must be replaced, each time it is used, by $\Gamma \succcurlyeq \alpha \Sigma$. This way, the operator inequality $(\Gamma + \beta I)^{-1} \preccurlyeq \frac{1}{\mu(1-\alpha)} (\Sigma_p + \beta I)^{-1}$ becomes $(\Gamma + \beta I)^{-1} \preccurlyeq \frac{1}{\alpha} (\Sigma_p + \beta I)^{-1}$ which explains the additional factor $\frac{1}{\alpha}$ in the final bound.

## B.4  Proof of Proposition 6

According to [Bach, 2022, Proposition 6] we have that the eigenvalues of $\Sigma_{\hat{p}}$ (resp $\Sigma_{\hat{q}}$) are the same than the ones of $1/n K_{\hat{p}}$; and we also have that for an eigenvalue $\lambda$ of $1/n K_{\hat{p}}$ with associated eigenvector $\boldsymbol{\alpha}$, the function $f = \sum_{i=1}^{n} \alpha_i \varphi(x_i)$ is an eigenvector of $\Sigma_{\hat{p}}$ associated to the same eigenvalue. Hence, denoting $(\lambda_i)_{i=1}^{n}$ the eigenvalues of $1/n K_{\hat{p}}$ and $\boldsymbol{a}^s$ the associated normalized eigenvectors, the first term in (2) writes:

$$\mathrm{Tr}(\Sigma_{\hat{p}} \log \Sigma_{\hat{p}}) = \mathrm{Tr}\left(\frac{1}{n}K_{\hat{p}} \log \frac{1}{n}K_{\hat{p}}\right) = \sum_{i=1}^{n} \lambda_i \log(\lambda_i).$$

We now turn to the second term in (2). Let $\phi_x = (\varphi(x_1), \dots, \varphi(x_n))^*$, $\phi_y = (\varphi(y_1), \dots, \varphi(y_m))^*$ and $\psi = \begin{pmatrix} \sqrt{\frac{\alpha}{n}} \phi_x \\ \sqrt{\frac{1-\alpha}{m}} \phi_y \end{pmatrix}$. We have $\Sigma_{\hat{p}} = \psi^T \begin{pmatrix} \frac{1}{\alpha}I & 0 \\ 0 & 0 \end{pmatrix} \psi$ and $\alpha\Sigma_{\hat{p}} + (1-\alpha)\Sigma_{\hat{q}} = \psi^T \psi$. We also remark that $\psi\psi^T = K$. Knowing that the operator $\psi^T\psi$ and the matrix $\psi\psi^T$ have the same spectrum, we will replace $\alpha\Sigma_{\hat{p}} + (1-\alpha)\Sigma_{\hat{q}}$ by $K$ in the expression of $\mathrm{KKL}_\alpha$, which we can do with the following lemma.

**Lemma 12.** *If $\psi \in \mathbb{R}^{d \times r}$ with $d > r$, $g : \mathbb{R}^+ \to \mathbb{R}$ and $\psi\psi^T$ is invertible, then*

$$g(\psi^T\psi) = \psi^T(\psi\psi^T)^{-\frac{1}{2}} g(\psi\psi^T)(\psi\psi^T)^{-\frac{1}{2}}\psi$$

*Proof.* Let $\psi = U Diag(S)V^T$ the singular value decomposition of $\psi$ with $U \in \mathbb{R}^{r \times r}$ and $V \in \mathbb{R}^{d \times d}$ orthonormal matrices. We have : $\psi\psi^T = U Diag(S^2)U^T$ and $\psi^T\psi = V Diag(S^2)V^T$, so, $g(\psi\psi^T) = U Diag(g(S^2))U^T$ and $g(\psi^T\psi) = V Diag(g(S^2))V^T$. And so, $g(\psi^T\psi) = VU^T g(\psi\psi^T)UV^T$. Noticing that $(\psi\psi^T)^{-\frac{1}{2}}\psi = UV^T$ concludes the proof. $\square$

By Lemma 12 we can write the second term in (2) as:

$$\mathrm{Tr}(\Sigma_{\hat{p}} \log(\alpha\Sigma_{\hat{p}} + (1-\alpha)\Sigma_{\hat{q}})) = \mathrm{Tr}\left(\psi^T \begin{pmatrix} \frac{1}{\alpha}I & 0 \\ 0 & 0 \end{pmatrix} \psi\psi^T(\psi\psi^T)^{-\frac{1}{2}} \log(\psi\psi^T)(\psi\psi^T)^{-\frac{1}{2}}\psi\right)$$

$$= \mathrm{Tr}\left(\begin{pmatrix} \frac{1}{\alpha}I & 0 \\ 0 & 0 \end{pmatrix} K^{\frac{1}{2}} \log(K) K^{\frac{1}{2}}\right)$$

$$= \mathrm{Tr}\left(\begin{pmatrix} \frac{1}{\alpha}I & 0 \\ 0 & 0 \end{pmatrix} K \log(K)\right),$$

where the last equality results from the fact that $K^{\frac{1}{2}}$ and $\log K$ commute because they have the same eigenbasis.

## B.5  Proof of Proposition 7

We write $\mathcal{F} = \mathcal{F}_1 + \mathcal{F}_2$ and derive the first variation of each functional in the next two lemmas. Then, we conclude on the first variation of $\mathcal{F}$.

**Lemma 13.** *Let $\hat{p}$ as defined in Proposition 6. The first variation of $\mathcal{F}_1 : \hat{p} \to \mathrm{Tr}(\Sigma_{\hat{p}} \log \Sigma_{\hat{p}})$ at $\hat{p}$ is, for $x \in \mathrm{supp}(\hat{p})$:*

$$\mathcal{F}_1'(\hat{p})(x) = \mathrm{Tr}(\varphi(x)\varphi(x)^*(I + \log \Sigma_{\hat{p}})),$$

*and $+\infty$ else.*

*Proof.* In this proof we use residual formula which is useful to derive spectral functions [1]. Indeed, we can write $\mathrm{Tr}(\Sigma_{\hat{p}} \log \Sigma_{\hat{p}}) = \sum_{i=1}^{n} f(\lambda_i(\Sigma_{\hat{p}}))$ with $f : \mathbb{C}\backslash\mathbb{R}^- \to \mathbb{C}, z \to z \log z$. Consider a perturbation $\xi \in \mathcal{P}(\mathbb{R}^d)$, $\varepsilon > 0$ and let $\Delta = \varepsilon\Sigma_\xi$. Let $z \in \mathbb{C}$. Using the linearity of $p \mapsto \Sigma_p$, we have

$$\mathrm{Tr}((\Sigma_{\hat{p}+\epsilon\xi}) \log(\Sigma_{\hat{p}+\epsilon\xi})) - \mathrm{Tr}(\Sigma_{\hat{p}} \log \Sigma_{\hat{p}}) = \mathrm{Tr}((\Sigma_{\hat{p}} + \Delta)\log(\Sigma_{\hat{p}} + \Delta)) - \mathrm{Tr}(\Sigma_{\hat{p}} \log \Sigma_{\hat{p}})$$

$$= \sum_{i=1}^{n} f(\lambda_i(\Sigma_{\hat{p}} + \Delta)) - f(\lambda_i(\Sigma_{\hat{p}})).$$

---

[1] See https://francisbach.com/cauchy-residue-formula/.

Let $\gamma$ be a closed directed contour in $\mathbb{C}\backslash\mathbb{R}^-$ which surrounds all the positive eigenvalues of $\Sigma_{\hat{p}}$ and $\Sigma_{\hat{p}} + \Delta$. We have

$$\sum_{i=1}^{n} f(\lambda_i(\Sigma_{\hat{p}} + \Delta)) - f(\lambda_i(\Sigma_{\hat{p}})) = \frac{1}{2i\pi} \oint_\gamma f(z)\operatorname{Tr}\left((zI - \Sigma_{\hat{p}} - \Delta)^{-1}\right) - f(z)\operatorname{Tr}\left((zI - \Sigma_{\hat{p}})^{-1}\right)dz$$

$$= \frac{1}{2i\pi} \oint_\gamma f(z)\operatorname{Tr}\left((zI - \Sigma_{\hat{p}} - \Delta)^{-1} - (zI - \Sigma_{\hat{p}})^{-1}\right)dz,$$

where

$$(zI - \Sigma_{\hat{p}} - \Delta)^{-1} - (zI - \Sigma_{\hat{p}})^{-1} = (zI - \Sigma_{\hat{p}})^{-1}\Delta(zI - \Sigma_{\hat{p}})^{-1} + o(\|\Delta\|_{op}).$$

Hence, denoting $\Sigma_{\hat{p}} = \sum_{i=1}^{n} \lambda_i f_i f_i^*$ the singular value decomposition of $\Sigma_{\hat{p}}$, we have

$$\operatorname{Tr}((\Sigma_{\hat{p}} + \Delta)\log(\Sigma_{\hat{p}} + \Delta)) - \operatorname{Tr}\Sigma_{\hat{p}}\log\Sigma_{\hat{p}}$$

$$= \frac{1}{2i\pi} \oint_\gamma f(z)\operatorname{Tr}\left((zI - \Sigma_{\hat{p}})^{-1}\Delta(zI - \Sigma_{\hat{p}})^{-1}\right)dz + o(\varepsilon)$$

$$= \frac{1}{2i\pi} \oint_\gamma f(z)\operatorname{Tr}\left(\sum_{i=1}^{n}\sum_{k=1}^{n} \frac{f_i f_i^* \Delta f_k f_k^*}{(z - \lambda_i)(z - \lambda_k)}\right)dz + o(\varepsilon)$$

$$= \frac{1}{2i\pi} \sum_{k=1}^{n} \oint_\gamma \frac{f(z)}{(z - \lambda_k)^2}dz\, \operatorname{Tr}(f_k^* f_k \Delta) + o(\varepsilon).$$

The residue of $h(z) = \frac{f(z)}{(z - \lambda_k)^2} = \frac{z\log z}{(z - \lambda_k)^2}$ at $\lambda_k$ is[2] $Res(h, \lambda_k) = 1 + \log\lambda_k$. Applying again the residue formula we have

$$\operatorname{Tr}((\Sigma_{\hat{p}} + \Delta)\log(\Sigma_{\hat{p}} + \Delta)) - \operatorname{Tr}(\Sigma_{\hat{p}}\log\Sigma_{\hat{p}}) = \sum_{k=1}^{n}(1 + \log\lambda_k)\operatorname{Tr}(f_k f_k^* \Delta) + o(\varepsilon)$$

$$= \operatorname{Tr}((I + \log\Sigma_{\hat{p}})\Delta) + o(\epsilon)$$

$$= 1 + \operatorname{Tr}(\log(\Sigma_{\hat{p}})\Delta) + o(\epsilon)$$

This concludes the proof by dividing the later quantity by $\epsilon$ and taking the limit as $\epsilon \to 0$. $\qquad\square$

**Lemma 14.** *Let $\hat{p}, \hat{q}$ as defined in Proposition 6. The first variation of $\mathcal{F}_2$ : $\hat{p} \to \operatorname{Tr}(\Sigma_{\hat{p}}\log(\alpha\Sigma_{\hat{p}} + (1 - \alpha)\Sigma_{\hat{q}}))$ at $\hat{p}$ is, for any $x \in \operatorname{supp}(\hat{p})$:*

$$\mathcal{F}_2'(\hat{p})(x) = \operatorname{Tr}\left(\log(\alpha\Sigma_{\hat{p}} + (1 - \alpha)\Sigma_{\hat{q}})\varphi(x)\varphi(x)^*\right)$$

$$+ \alpha\operatorname{Tr}\left(\left(\sum_{j=1}^{n+m} \frac{h_j h_j^* \Sigma_{\hat{p}} h_j h_j^*}{\eta_j} + \sum_{j\neq k} \frac{\log\eta_j - \log\eta_k}{\eta_j - \eta_k} h_j h_j^* \Sigma_{\hat{p}} h_k h_k^*\right)\varphi(x)\varphi(x)^*\right), \quad (31)$$

*where $(\eta_j, h_j)_{i=1}^{n+m}$ are the eigenvalues and eigenvectors of $\alpha\Sigma_{\hat{p}} + (1 - \alpha)\Sigma_{\hat{q}}$.*

*Proof.* Denote $\hat{\Gamma} = (1 - \alpha)\Sigma_{\hat{q}} + \alpha\Sigma_{\hat{p}}$. As for Lemma 13, let $\Delta = \varepsilon\Sigma_\xi$. We have:

$$\operatorname{Tr}(\Sigma_{\hat{p}+\Delta}\log(\alpha\Sigma_{\hat{p}+\Delta} + (1 - \alpha)\Sigma_{\hat{q}})) = \operatorname{Tr}(\Sigma_{\hat{p}} + \Delta)\log\left(\hat{\Gamma} + \alpha\Delta\right) - \operatorname{Tr}(\Sigma_{\hat{p}})\log\hat{\Gamma}$$

$$= \operatorname{Tr}\left((\Sigma_{\hat{p}} + \Delta)\log\left(\hat{\Gamma} + \alpha\Delta\right)\right) - \operatorname{Tr}\left((\Sigma_{\hat{p}} + \Delta)\log\hat{\Gamma}\right) + \operatorname{Tr}\left((\Sigma_{\hat{p}} + \Delta)\log\hat{\Gamma}\right) - \operatorname{Tr}\left(\Sigma_{\hat{p}}\log\hat{\Gamma}\right)$$

$$= \operatorname{Tr}\left(\Sigma_{\hat{p}}(\log\left(\hat{\Gamma} + \alpha\Delta\right) - \log\hat{\Gamma})\right) + \operatorname{Tr}\left(\Delta\log\hat{\Gamma}\right).$$

The second term on the r.h.s. is already linear in $\Delta$ as desired, hence we focus on the first one. Using a singular value decomposition of $\hat{\Gamma} + \alpha\Delta$ and $\hat{\Gamma}$ we write: :

$$\operatorname{Tr}\left(\Sigma_{\hat{p}}(\log\left(\hat{\Gamma} + \alpha\Delta\right) - \log\hat{\Gamma})\right) =$$

$$\operatorname{Tr}\left(\Sigma_{\hat{p}}\left(\sum_{j=1}^{n+m} \log\eta_j(\hat{\Gamma} + \alpha\Delta)h_j' h_j'^*\right)\right) - \operatorname{Tr}\left(\Sigma_{\hat{p}}\left(\sum_{j=1}^{n+m} \log\eta_j(\hat{\Gamma})h_j h_j^*\right)\right),$$

---

[2]Using that if $h(z) = \frac{f(z)}{(z - \lambda)^2}$, then $Res(h, \lambda) = f''(z)$ where $f(z) = z\log(z)$. Recall that $Res(h, \lambda) = \frac{1}{2i\pi} \oint_\gamma h(z)dz$ where $\gamma$ is a contour circling strictly $\lambda$.

where $(h'_j)_j$ are the eigenvectors of positive eigenvalues of $\hat{\Gamma} + \alpha\Delta$. Let $\gamma$ a loop surrounding all the eigenvalues $\eta_j(\hat{\Gamma} + \alpha\Delta)$ and $\eta_j(\hat{\Gamma})$, then,

$$\sum_{j=1}^{n+m} \log \eta_j(\hat{\Gamma} + \alpha\Delta)h'_j h'^*_j = \frac{1}{2i\pi} \oint_\gamma \log(z)(zI - \hat{\Gamma} - \alpha\Delta)^{-1}dz$$

$$\text{and} \sum_{j=1}^{n+m} \log \eta_j(\hat{\Gamma})h_j h^*_j = \frac{1}{2i\pi} \oint_\gamma \log(z)(zI - \hat{\Gamma})^{-1}dz.$$

Moreover, we have $(zI - \hat{\Gamma} - \alpha\Delta)^{-1} - (zI - \hat{\Gamma})^{-1} = (zI - \hat{\Gamma})^{-1}\alpha\Delta(zI - \hat{\Gamma})^{-1} + o(\varepsilon)$. Hence,

$$\text{Tr}\Big(\Sigma_{\hat{p}}(\log\big(\hat{\Gamma} + \alpha\Delta\big) - \log\hat{\Gamma})\Big) = \frac{1}{2i\pi} \oint_\gamma \text{Tr}\Big(\Sigma_{\hat{p}}\log(z)(zI - \hat{\Gamma})^{-1}\alpha\Delta(zI - \hat{\Gamma})^{-1}\Big)dz + o(\varepsilon)$$

$$= \frac{\alpha}{2i\pi} \oint_\gamma \log(z)\,\text{Tr}\left(\Sigma_{\hat{p}}\left(\sum_{j,k=1}^{n+m} \frac{h_j h^*_j \Delta h_k h^*_k}{(z - \eta_j)(z - \eta_k)}\right)\right)dz$$

$$= \frac{\alpha}{2i\pi} \left(\sum_{j,k=1}^{n+m} \oint_\gamma \frac{\log(z)}{(z - \eta_j)(z - \eta_k)}dz\,\text{Tr}\big(\Sigma_{\hat{p}}h_j h^*_j \Delta h_k h^*_k\big)\right).$$

With the residue theorem, for $j \neq k$, $\oint_\gamma \frac{\log(z)}{(z-\eta_j)(z-\eta_k)}dz = 2i\pi\left(\frac{\log\eta_j}{(\eta_j-\eta_k)} + \frac{\log\eta_k}{(\eta_k-\eta_j)}\right) = 2i\pi\frac{\log\eta_j - \log\eta_k}{\eta_j - \eta_k}$, and for $k = j$, $\oint_\gamma \frac{\log(z)}{(z-\eta_j)^2}dz = \frac{2i\pi}{\eta_j}$. We then have:

$$\text{Tr}\Big(\Sigma_{\hat{p}}(\log\big(\hat{\Gamma} + \alpha\Delta\big) - \log\hat{\Gamma})\Big) = \alpha\sum_{j=1}^{n+m} \frac{1}{\eta_j}\text{Tr}\big(h_j h^*_j \Sigma_{\hat{p}}h_j h^*_j \Delta\big)$$

$$+ \alpha\sum_{j\neq k}^{n+m} \frac{\log\eta_j - \log\eta_k}{\eta_j - \eta_k}\text{Tr}\big(h_k h^*_k \Sigma_{\hat{p}}h_j h^*_j \Delta\big) + o(\varepsilon).$$

We note that if $\Sigma_{\hat{p}}$ an $\Sigma_{\hat{q}}$ were diagonalizable in the same eigenbasis, then the previous quantity would be equal to $\alpha\sum_{j=1}^{n+m} \frac{\lambda_j}{\eta_j}\text{Tr}\big(h_j h^*_j \Delta\big) = \text{Tr}\,\Sigma_p \Gamma^\dagger \Delta$ where $\Gamma^\dagger$ is the pseudo inverse of $\Gamma$. We conclude again dividing the latter quantity by $\epsilon$ and considering its limit as $\epsilon \to 0$. $\qquad\square$

We can now write the matrix expression for the first variation of $\mathcal{F}$ using Lemma 12. We remind that $\phi_x = \begin{pmatrix} \varphi(x_1)^* \\ .. \\ \varphi(x_n)^* \end{pmatrix}$ (resp $\phi_y$) and $\psi = \begin{pmatrix} \sqrt{\frac{\alpha}{n}}\phi_x \\ \sqrt{\frac{1-\alpha}{m}}\phi_y. \end{pmatrix}$, and that $\psi^T\psi = \alpha\Sigma_{\hat{p}} + (1 - \alpha)\Sigma_{\hat{q}}$ and $\psi\psi^T = K$. We remark that $\Sigma_{\hat{p}} = \frac{1}{\sqrt{n}}\phi_x^T \frac{1}{\sqrt{n}}\phi_x$ and $\phi_x\phi_x^T = \frac{1}{n}K_{\hat{p}}$. By Lemma 12, we have

$$\text{Tr}(\log(\Sigma_{\hat{p}})\varphi(x)\varphi(x)^*) = \text{Tr}\left(\frac{1}{n}\phi_x^T\left(\frac{1}{n}K_{\hat{p}}\right)^{-\frac{1}{2}}\log\left(\frac{1}{n}K_{\hat{p}}\right)\left(\frac{1}{n}K_{\hat{p}}\right)^{-\frac{1}{2}}\phi_x\varphi(x)\varphi(x)^*\right)$$

$$= S(x)^T g(K_{\hat{p}})S(x)$$

since $S(x) = \phi_x\varphi(x)$. We show the same way that $\text{Tr}\log(\alpha\Sigma_{\hat{p}} + (1 - \alpha)\Sigma_{\hat{q}})\varphi(x)\varphi(x)^* = T(x)^T g(K)T(x)$.

For the third term in the first variation of $\mathcal{F} = \mathcal{F}_1 + \mathcal{F}_2$, i.e. the second one in Equation (31), our goal is to rewrite $\text{Tr}\big(h_k h^*_k \Sigma_{\hat{p}}h_j h^*_j \Delta\big)$ in terms of matrices. We have $h_k = \psi^T c_k/\|\psi^T c_k\|$ (idem j)

where $c_k$ is an eigenvector of $K$ of eigenvalue $\eta_k$, and $\|\psi^T c_k\|^2 = K c_k = \eta_k$. Hence,

$$
\begin{aligned}
\mathrm{Tr}\big(h_k h_k^* \Sigma_{\hat{p}} h_j h_j^* \Delta\big) &= \mathrm{Tr}\bigg( \psi^T c_j c_j^T \psi \psi^T \begin{pmatrix} \frac{1}{\alpha}I & 0 \\ 0 & 0 \end{pmatrix} \psi \psi^T c_k c_k^T \psi \Delta \bigg) / (\eta_k \eta_j) \\
&= \mathrm{Tr}\bigg( c_j c_j^T K \begin{pmatrix} \frac{1}{\alpha}I & 0 \\ 0 & 0 \end{pmatrix} K c_k c_k^T \psi \Delta \psi^T \bigg) / (\eta_k \eta_j) \\
&= \mathrm{Tr}\bigg( c_j c_j^T \begin{pmatrix} \frac{1}{\alpha}I & 0 \\ 0 & 0 \end{pmatrix} c_k c_k^T \psi \Delta \psi^T \bigg) \\
&= \varepsilon \int \mathrm{Tr}\bigg( c_j c_j^T \begin{pmatrix} \frac{1}{\alpha}I & 0 \\ 0 & 0 \end{pmatrix} c_k c_k^T \psi \varphi(x) \varphi(x)^* \psi^T \bigg) d\xi(x).
\end{aligned}
$$

We have $\psi \varphi(x) \varphi(x)^* \psi^T = V(x) V(x)^T$ where $V(x) = \big( \frac{\alpha}{n} k(x, x_1), \dots, \frac{1-\alpha}{m} k(x, y_1) \big)^T$, and if we note $c_j = (a_j, b_j)^T$:

$$
c_j c_j^T \begin{pmatrix} \frac{1}{\alpha}I & 0 \\ 0 & 0 \end{pmatrix} c_k c_k^T = \frac{\langle a_j, a_k \rangle}{\alpha} c_j c_k^T.
$$

Finally,

$$
\mathrm{Tr}\Big( \Sigma_{\hat{p}}(\log\big( \hat{\Gamma} + \alpha\Delta \big) - \log \hat{\Gamma}) \Big) =
$$
$$
\varepsilon \int \mathrm{Tr}\left( \sum_{j=1}^{n+m} \frac{\|a_j\|^2}{\eta_j} c_j c_j^T + \sum_{j \neq k} \frac{\log \eta_j - \log \eta_k}{\eta_j - \eta_k} \langle a_j, a_k \rangle c_j c_k^T \right) V(x) V(x)^T. \quad (32)
$$

## C   Additional Experiments

**Skewness and concentration of the** KKL**.**   In these examples, we plot $\mathrm{KKL}_\alpha(\hat{p}\|\hat{q})$ as the number of $n = m$ samples of two distributions $p, q$ increases. In Figure 4 we plot the KKL for two Gaussians by varying the dimension $d$. It can be seen that the larger $d$ is, the less KKL oscillates. We can also remark that the value of KKL increases with the dimension, reflecting the effect of the constants of Proposition 4. Figure 5 is the same experiment as in the main text, except that the dimension is 2. We can also see here that $\mathrm{KKL}_\alpha$ is monotone in $\alpha$. We can also see that convergence to the value of KKL in population is faster in this case than for $d = 10$. The third experiment in Figure 6 is in dimension 1 and the distribution of $q$ is an exponential distribution with parameter $\lambda = 1$, while $p$ is a Gaussian distribution. We can notice a few points, such as for example that the values taken by KKL are smaller and that it varies less with $\alpha$ than in the case of Figure 5.

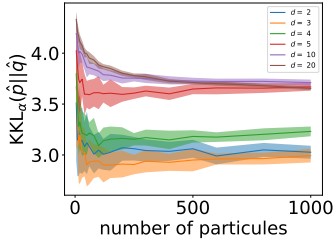 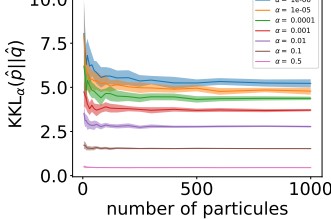 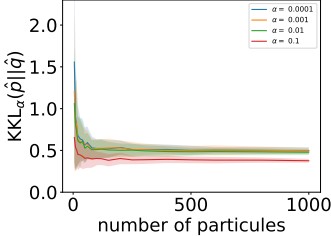

Figure 4: $\alpha = 0.01$, $p, q$ Gaussians, $\sigma$ is the square of the mean of distances between $\hat{p}$ and $\hat{q}$.

Figure 5: $\alpha = 0.1$, $\sigma = 2$.

Figure 6: $p \sim \mathcal{N}(1, 1)$, $q \sim \mathcal{E}(1)$, $\alpha = 0.1$, $\sigma = 2$.

**Sampling with KKL gradient descent on Gaussians and mixtures of Gaussians.**   We are interested in sampling on Gaussians or mixtures of 2 Gaussians by varying the dimension. Figure 7 and Figure 8 show the evolution of the KKL value during the gradient descent of different dimensions $d$, starting with a Gaussian $p$ and taking $q$ to be a mixture of 2 Gaussians for Figure 7 and $p$ and $q$ Gaussians distributions for Figure 8. In Figure 7 and Figure 8, the stepsize

$h = \frac{1}{n} \left( \sum_{i,j} \|x_i - y_j\|^2 \right)^{1/2} n^{-1/(d+4)}$. For each $d$, we report the average and error bars of our results for 20 runs, varying the samples drawn from the initialization and the thick lines represents the average value. We can see that in both cases the convergence is faster for small values of $d$. On Figure 9 we observe the evolution of $W_2(\hat{p}\|\hat{q})$, the 2 Wasserstein distance, during gradient descent in dimension $d = 10$ for various parameters $\alpha$. The distribution $p$ and $q$ are respectively a Gaussian and a mixture of 2 Gaussians. The values of $W_2$ at each iteration $t$ is computed as the mean of $W_2(\hat{p}, \hat{q})$ on 10 runs of the gradient descent where for each the mean of $p$ is drawn at random. We can see that if the $\alpha$ value is too high, then convergence in 2-Wasserstein is slower, whereas if it is too small, convergence is faster at the beginning, but does not lead to an optimal value in Wasserstein distance at the end of the algorithm.

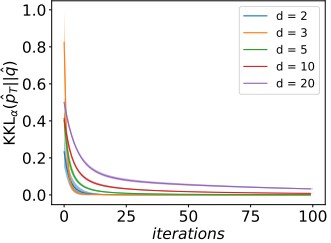 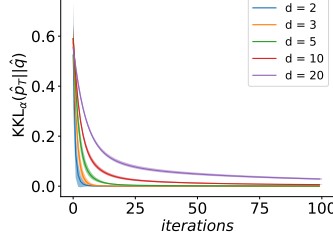 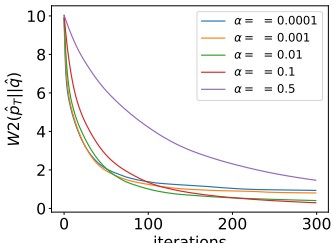

Figure 7: $\alpha = 0.01$, $p$ is a Gaussian distribution and $q$ a mixture of 2 Gaussians, $\sigma$ is the square of the mean of distances between $\hat{p}$ and $\hat{q}$.

Figure 8: $\alpha = 0.01$, $p$ and $q$ are Gaussians. Bandwidth $\sigma$ is the mean of the square distances between $\hat{p}$ and $\hat{q}$ points.

Figure 9: $\sigma = 10$, $h = 5$, $p$ is Gaussian and $q$ is a mixture of 2 Gaussians.

**3 rings.** Appendix C compares the evolution of the gradient flows of MMD, Kale and $\mathrm{KKL}_\alpha$ in terms of Wassertsein distance and Energy distance in the case where optimisation of KKL is done with L-BFGS linesearch in Figure 2. We observe that both Kale and KKL seem to converge towards 0 in terms of energy distance and Wasserstein distance but $\mathrm{KKL}_\alpha$ is faster to converge, in term of number of iterations than Kale. The MMD flow decreases the energy distance but does not converge to 0 in 2-Wasserstein distance, unlike Kale and KKL, reflecting the fact that some particles are not supported on the target support. The bandwidth of $k$ is fixed at $\sigma = 0.1$ for Kale and MMD and at $\sigma = 0.3$ for KKL. In Figure 13 this time we repeat the experiment but for a simple gradient descent for KKL with constant step $h = 0.01$. We see that in this case the speed of convergence in terms of iterations for KKL is slower than for Kale (there are only about 100 iterations necessary for Kale and MMD and 300 for KKL) but it ends up obtaining (see Figure 12), in terms of Wassertsein distance, a similar limit. On the other hand, the execution time of the gradient descent for 300 iterations of KKL is about the same as for Kale and MMD for 100 iterations.

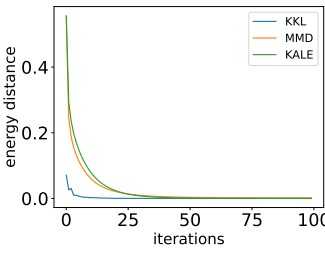 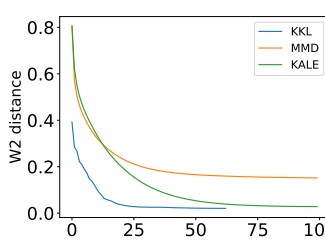 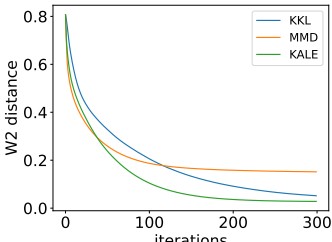

Figure 10: L-BFGS, $\sigma = 0.3$, $\alpha = 0.01$

Figure 11: L-BFGS, $\sigma = 0.3$, $\alpha = 0.01$

Figure 12: Constant step size $h = 0.01$, $\sigma = 0.3$, $\alpha = 0.001$

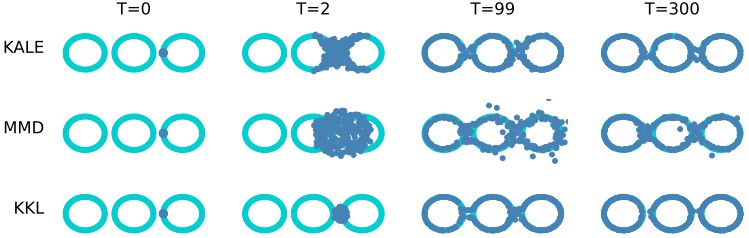

Figure 13

