# OpenReview forum: "Statistical and Geometrical properties of the Kernel Kullback-Leibler divergence"
_NeurIPS.cc/2024/Conference — NeurIPS 2024 poster_

### Official Review · Reviewer_6zq3 · 2024-06-19

**Soundness:** 4
**Presentation:** 3
**Contribution:** 3
**Rating:** 6
**Confidence:** 4

**Summary:**

This paper proposes the regularized kernel KL divergences. With the kernelization, the authors derive closed form formula and makes it computationally cheap to optimize using gradient methods. Theoretically, a new finite sample estimation convergence bound is derived. Numerical simulations suggest this proposed divergences are not only easy to implement, but also suitable for sampling complicated distributions.

**Strengths:**

1. The paper is well-written and easy to follow.

2. Finite sample approximation bound is derived.

3. Numerical simulations demonstrate good performance over other related divergences.

**Weaknesses:**

1. Proposition 1 still assumes $p$ is absolutely continuous with respect to $q$.

2. The bound in Proposition 3 becomes trivial since it approaches $\infty$ as $\alpha\to 1$.

**Questions:**

1. Typo at line 301: ''For small values of the regularization parameter'' $\to$ ''For large values of the regularization parameter'.

2. Can the authors comment on how the performance of KKL compares with that of skewed KL divergence which is kernel-free and not restricted by a kernel, though the later one can be computationally more expensive?

3. You mention your algorithm is robust to the choice of $\alpha$. But I was wondering when $\alpha$ is close to 1, as suggested in Figure 1 even for $\alpha=0.5$, the value of the divergence is always small, so it will be hard to distinguish between different distributions. Can you comment on this?

4. Why does Figure 2(a) begin with different particles across different methods at $T=0$?

**Limitations:**

Yes.

---

> ### Author Rebuttal · Authors · 2024-08-04
>
> We thank the reviewer for careful reading the paper and  for his relevant suggestions. We have addressed each of your comments below. Please don't hesitate to let us know if you have any further questions.
>
> - Weakness: "Proposition 1 still assumes $p$
>  is absolutely continuous with respect to $q$
> ."
>
> **Reply:** Thank you for spotting the typo,  please see the general comment.
>
> - Weakness : "The bound in Proposition 3 becomes trivial since it approaches $\infty$ as $\alpha \rightarrow 1$."
>
> **Reply:* This is true that the bound becomes trivial as $\alpha$ goes to 1.
> However, our objective in calculating this upper bound was to prove that when $\alpha$ tends to 0, the regularized KKL converges to the true KKL. We are therefore mainly interested in this bound for small values of alpha.
>
> - Question: "Can the authors comment on how the performance of KKL compares with that of skewed KL divergence which is kernel-free and not restricted by a kernel, though the later one can be computationally more expensive?"
>
> **Reply** The skewed KL is well defined even if $p$ is not absolutely continuous with respect to $q$, but it seems to us that if $p$ and $q$ are atomic measures, and that the support of $p$ will move (for instance in the gradient flow setting as in our paper), then one would need to use kernel-based density estimates at each iteration. This could be an interesting idea to compare with the regularized KKL. We originally did not include it, because from [1], it is known that the KKL is a better approximation of the KL than a KL between smooth estimates of $p,q$ for a specific smoothing kernel, see Equation (8) in [1] which states that   $KL(\tilde{p}||\tilde{q}) \le KKL(p||q) \le KL(p||q) $, where $\tilde{p}$ and $\tilde{q}$ are smoothed version of $p$ and $q$. However, this is a simple baseline to implement that we will include in our experiments in the revised manuscript.
>
> - Question: "You mention your algorithm is robust to the choice of $\alpha$. But I was wondering when $\alpha$ is close to 1, as suggested in Figure 1 even for $\alpha = 0.5$, the value of the divergence is always small, so it will be hard to distinguish between different distributions. Can you comment on this?"
>
> **Reply:** We agree with the reviewer, and this is a problem that would appear with the regular KL - see section A.2 that recalls the monotony of skewed KL with respect to the skewness parameter $\alpha$. One of our contributions  is to show that the skewed KKL verifies the same property (see Proposition 2). This is the reason why we tend to use small values of alpha in the rest of the experiments, see for instance Appendix C describing the hyperparameters for different experiments, $\alpha$ is typically of order $10^{-2}$ or $10^{-3}$. For larger values of $\alpha$, i.e. $\alpha=0.5$, an alternative idea would be to consider a "Jensen-Shannon" (JS) version of regularized KKL. Indeed alpha= 0.5 in the skewed (standard- KL can be used to design a JS divergence (see Remark 1) and the second term is used to balance this phenomenon. We think that this is an interesting idea, and will include illustrative experiments with a Jensen-Shannon version of our KKL divergence for the same experimental settings considered in our paper.
>
> - Question : "Why does Figure 2(a) begin with different particles across different methods at $T=0$ ?"
>
> **Reply:** You indeed spotted a mistake, for KKL, we plotted at T= 1 instead of T = 0 and this is why the initialisation looks different. In the submitted version, we corrected this mistake. You can see the corrected figure in a pdf in the general comment.

---

> > ### Comment · Reviewer_6zq3 · 2024-08-09
> > **Reply to authors**
> >
> > I thank the authors for the detailed response and I will keep my score.

---

### Official Review · Reviewer_ffGL · 2024-07-10

**Soundness:** 3
**Presentation:** 3
**Contribution:** 3
**Rating:** 5
**Confidence:** 4

**Summary:**

The authors generalize and analyze the kernel Kullback-Leibler divergence introduced by Francis Bach a few years ago. The main contributions are 1) use a skew divergence (like in JS) tinstead of KL  consider non-absolutely continuous divergences, 2) provide an explicit formula of the divergence for discrete measure which allows a direct implementation for data, 3) provide refined finite sample guarantees. The divergence is used then to build a Wasserstein gradient flow and generative modeling. Some low-dimensional numerical examples are provided.

**Strengths:**

The work extend the original work by F. Bach in several aspects. In particular the explicit formula of the divergence for discrete measures is a nice and very useful results which wiil come handy  in applications. The paper is very clearly and nicely written and the proof looked sound to the reviewer.

**Weaknesses:**

The reviewer find the applications somewhat underwhelming. The gradient flows built via kernel methods do not seem to perform particularly well, perhaps due to the well-known lack of expressivity. One might have hope that the covariance embeddings in these new divergence would have helped compared to MMD based gradient flow but the authors experiments do not make that case.  Kernel methods have the advantages that provide explicit formula for the velocity vector field and one would hope that this allow to consider high-dimensional problem but no such experiment have been provided so it is not clear if the proposed model scales up.

**Questions:**

1) Is there a reasonable expectation that their flows to scale up to high dimension? Or what is the obstacle?

2) It seems to the reviewer that Wasserstein gradient flow based on kernel methods do not seem to perform well compared to gradient flows using neural network architecture.  This occurs in the case of SVGD but also for plain Wasserstein gradient flows. For example the gradient flows in Gu et al (https://arxiv.org/abs/2210.17230) based on Lipschitz regularization of KL-divergence are comparable in spirit to the MMD of Kernel-KL flows considered here.  Is the reviewer mistaken? Can the authors provide some insights on these issues.

**Limitations:**

The limitations of the work are properly addressed.

---

> ### Author Rebuttal · Authors · 2024-08-04
>
> Thank you for your feedback and time. We have addressed each of your comment below. If you have any further question, please don't hesitate to let us know.
>
> - Weakness: " The gradient flows built via kernel methods do not seem to perform particularly well, perhaps due to the well-known lack of expressivity. One might have hope that the covariance embeddings in these new divergence would have helped compared to MMD based gradient flow but the authors experiments do not make that case."
>
> **Reply** We respectfully strongly disagree. We see (visually) on Figures 2.a and 2.b on page 9 that the KKL flow converges to a better particle configuration than the MMD one, even after carefully tuning all the hyperparameters, including for MMD (that is notoriously hard to optimize, see [1]). This is also seen when reporting the convergence with respect to different metrics in the Appendix C (see paragraph "3 rings" starting l683 and Figure 10). We do not understand why the reviewer claims the converse statement. Moreover, it has also been shown the paper cited above [1], that MMD gradient flow alone does not converge well even on simple experiments, e.g. between Gaussians as in Figure 2 in [1]. KKL flow also works better than KALE in the case where L-BFGS method is used, a method which cannot be used for the KALE flow as it doesn't have a closed form expression for its loss and gradient (that are requirements for L-BFGS). If the reviewer disagree or has questions on the experiments, please let us know so that we can clarify.
>
>
> - Question "Is there a reasonable expectation that their flows to scale up to high dimension? Or what is the obstacle?"
>
> **Reply**: Notice that in Appendix C, we conduct experiments on mixture of Gaussians up to dimension 10, see paragraph starting l669 and figures at the top of p26).  We acknowledge that we did not try to use KKL in higher dimensions such as the one of images or in a more challenging generative modeling setting. A first step would be to work on reducing the computational complexity of KKL (which is $O((n+m)^3)$ as discussed l251 wrt to the number of samples, that should be multiplied by $d$ which is the kernel computation cost) to work on high-dimensional and large datasets. We leave this study for future work.
>
> - Question "It seems to the reviewer that Wasserstein gradient flow based on kernel methods do not seem to perform well compared to gradient flows using neural network architecture. This occurs in the case of SVGD but also for plain Wasserstein gradient flows. For example the gradient flows in [2] based on Lipschitz regularization of KL-divergence are comparable in spirit to the MMD of Kernel-KL flows considered here. Is the reviewer mistaken? Can the authors provide some insights on these issues."
>
> **Reply:** There might be a slight confusion here, but let us know please if we misunderstood. There exists 2 types of kernel-based approximations of the Kullback-Leibler divergence that were proposed in the literature: (a) the one we propose here, based on kernel covariance embeddings  and (b) the "variational one" (see Equation (12) in our paper).
> The paper you are referring to is analog to the second type; the difference being that the variational family there is defined by neural networks. Regarding the performance of this divergence compared to ours, there are two ingredients at play: (1) first, the type of approximation, and (2) second the parametric family.
> Regarding (1), it is not clear how our kernel-based approximation of the standard KL (i.e. type (a)) is "better" than the variational approximation such as KALE (i.e. type (b)).
> Our work, through numerical experiments, is a first tentative for comparing these different techniques, through comparing KKL with Kale.
> Then, the choice of parametric family is crucial indeed. In the reference mentioned by the reviewer, the variational family is approximated through neural networks instead of a Gaussian RKHS as in KALE. As explained in Section 1 therein, when the variational family is an RKHS ball, these approximated divergences interpolate between MMD and KL divergence, while if it is the class of 1-Lipschitz functions, it interpolates between Wasserstein-1 (which has much better topological and geometrical properties than MMD) and KL. In [2], the space of 1-Lipschitz functions is approximated by a space of neural network.
> A possible (and fair) comparison in our framework (i.e. type (a) using the KKL) may be to learn a kernel feature map with neural networks before applying KKL formula - instead of considering directly Gaussian kernels as in our paper. We think this is an interesting but deep question that requires further investigation.
>
>
>
>
> [1] Arbel, Korba, Salim, Gretton. Maximum Mean Discrepancy Gradient Flow, Neurips, 2019.
>
> [2] Gu, Birmpa, Pantazis, Ret-Belley, A. Katsoulakis, Lipschitz-Regularized Gradient Flows and Generative Particle Algorithms for High dimensional Scarce Data, 2023

---

> > ### Comment · Reviewer_ffGL · 2024-08-12
> >
> > Thank you for the detailed answer and I will keep my score.

---

### Official Review · Reviewer_19Wg · 2024-07-14

**Soundness:** 2
**Presentation:** 3
**Contribution:** 2
**Rating:** 5
**Confidence:** 4

**Summary:**

Comparing probability measures is a fundamental task in many statistical and machine learning problems. One of the metrics that has been ubiquitously used is Kullback-Leibler (KL) divergence. KL contrasts the information contained in two probability distributions. Statistical learning using KL divergence involves density ratios and needs the fact that the supports of probability measures in question should be not disjoint.

A kernel variant of KL called kernel KL (KLL) divergence was proposed in Bach (2022, IEEE Trans. Info.). KLL compares probability measures through covariance operators in an RKHS space.  As the standard KL, the KLL shares the limit of inability to compare measures with disjoint supports. This paper proposes a regularized variant of KLL that guarantees calculating divergence for all distributions. The authors derive bounds that quantify the deviation of regularized KLL to the original KLL.  they further propose a closed form for KLL for empirical measures. Numerical experiments are conducted on simulated data to corroborate the theoretical results.

**Strengths:**

- Proposing $KKL_\alpha$ a regularized version of standard KLL, that allows calculating divergence for all distributions.
- Presenting statistical properties of $KKL_\alpha$.
- Providing a closed for KLL between empirical measures.
- Plugging $KKL_\alpha$ in a Wasserstein gradient flow learning.

**Weaknesses:**

**Weakness and Questions**

- The decreasing property  of $KLL_\alpha$ in Proposition 2 needs the condition that $p$ is absolutely continuous with respect to $q$. However, in the definition of $KLL_\alpha$  (L144-L145) there is no need for this absolutely continuous. My question does this property still valid without $p<<q$ ? The same remark for the result in Proposition 4.
- In Proposition 4, the definition of $\varphi(x)$ needs to be recalled.
- In Proposition 4, could you please motivate the condition On the Radon-Nikodym derivative to be bound above by $1/\mu$.
- To facilitate the lecture on the bound given in Equation (5), I think it will be better to write its order.
- In Proposition 5, I didn't understand the definition of the gram kernels $K_x, K_y, K_{xy}$. Do these operators design the covariance operator or other more general kernels?

**Questions:**

**Minor Typos**

- L108: the notation $p$ is used for denoting probability measure and the dimension $\mathbb{R}^p$.
- L368: there is no caption of the figure and for (b) Shape transfer, the iteration $T=99$ is the same.

---

> ### Author Rebuttal · Authors · 2024-08-04
>
> Thank you for your careful reading and relevant suggestions. We have addressed your comments point-by-point below. Please don't hesitate to let us know if you have any further questions.
>
>
> - Question: "The decreasing property of $KKL_{\alpha}$ in Proposition 2 needs the condition that $p$ is absolutely continuous with respect to $q$. However, in the definition of
>  $KKL_{\alpha}$ (L144-L145) there is no need for this absolutely continuous. My question does this property still valid without $p \ll q$ ? The same remark for the result in Proposition 4."
>
> **Reply:** See the first point of general comment for the reply concerning Proposition 2.
>
> Concerning Proposition 4, it is an interesting and  important point that you bring up. We chose to make the assumption $p\ll q$ similarly than in Proposition 3, which shows the convergence of the regularized KKL to the true KKL in the case where $p \ll q$ as the regularization parameter $\alpha \to 0$. Under this common assumption, we can thus use Proposition 3 and 4 simultaneously to
> control the deviation of the regularized KKL on empirical distributions versus the unregularized KKL between $p$ and $q$. Additionally, under this assumption,  the bound in Proposition 4 scales with respect to the regularization parameter $\alpha$ as $\mathcal{O}(1/\alpha)$.
>
> The reviewer is right that this assumption is not the minimal assumption one could work with to obtain guarantees on the finite-sample approximation guarantees as we derive in Proposition 4. Indeed, it is possible to derive a similar rate of convergence without the assumption of absolute continuity, i.e. a bound that scales similarly with the number of samples $n,m$, by slightly modifying the proof (see details below). However, in this case, the bound would scale with respect to the regularization parameter as $\mathcal{O}(1/\alpha^2)$. We chose to propose only the first one in the paper as it scales better with $\alpha$ when $\alpha$ is small, but we realize thanks to the comment of the reviewer that this is an important point, and we will provide the alternative one.
>
> We provide in this paragraph more details about why the scale with respect to $\alpha$ changes depending on the hypothesis. In the proof, we repeatedly upper bound different terms by a quantity depending on $C(\beta) = \sup_x \langle \varphi(x), (\Sigma_p + \beta I)^{-1} \varphi(x) \rangle$, using  by taking advantage of the assumption that $c = \int C(\beta)^2 d\beta$ is finite. Also, we regularly have to deal with terms in which the operator $(N + \beta I)^{-1}$ appears, where $N = \alpha \Sigma_p + (1-\alpha) \Sigma_q$. Under the assumption $p\ll q$ and $dp/dq \le 1/\mu$, by operator inequality, the operator $(N + \beta I)^{-1}$ is upper bounded by $\frac{1}{(1- \alpha)\mu}(\Sigma_p + \beta I)^{-1}$, using the inequality $N \succeq (1-\alpha) \Sigma_q \succeq (1-\alpha) \mu \Sigma_p$. This will result ultimately in only one $\alpha$ in the denominator of our bound, i.e. the scale $\mathcal{O}(1/\alpha)$ mentioned above. Alternatively, it is possible to bound $(N + \beta I)^{-1}$ by $\frac{1}{\alpha}(\Sigma_p + \beta I)^{-1}$ using the simpler $N \succeq \alpha \Sigma_p$ and this does not require absolute continuity of $p\ll q$. However, this ultimately results in an additional factor $\frac1{\alpha}$ in our bound compared to the previous case, hence the scale $\mathcal{O}(1/\alpha^2)$, that is less favorable when the regularization parameter $\alpha$ is small. The corresponding bound is this one:
>
> $$\mathbb{E} | KKL_{\alpha}(\hat{p}||\hat{q}) - KKL_{\alpha}(p||q) | \leqslant \frac{32}{\alpha \sqrt{m \wedge n}} (2 \sqrt{c} + \log n) + \frac{2}{m \wedge n} \left(\frac{1}{\alpha} +   \frac{c(26 \log n)^2}{\alpha^2} (1 + \frac{n}{m \wedge m})\right).$$
>
>
> Thank you for pointing out this essential point, which we will clarify in the paper by giving an alternative proof and an alternative bound.
>
>
>
> - Question: "In Proposition 4, the definition of $\varphi(x)$ needs to be recalled."
>
> **Reply:** Thanks for the suggestion, we will indeed recall that $\varphi(x)$ is the feature map of $x \in \mathbb{R}^d$ in the RKHS $\mathcal{H}$.
>
>
> - Question: "In Proposition 4, could you please motivate the condition on the Radon-Nikodym derivative to be bounded above by $\frac1{\mu}$."
> .
>
> **Reply:**  see answer above; in summary it enables us to get a tighter bound with respect to the regularization parameter $\alpha$ when it gets smaller.
>
>
>
> - Question: "To facilitate the lecture on the bound given in Equation (5), I think it will be better to write its order."
>
> **Reply:** Thanks for the suggestion, we already worked on this post submission and think that this is an excellent suggestion to present our results in this simplified manner. A simplified bound is provided in the general comment to all the reviewers.
>
> - Question: "In Proposition 5, I didn't understand the definition of the gram kernels $K_x$, $K_y$, $K_{xy}$. Do these operators design the covariance operator or other more general kernels?
>
>
> **Reply:** Our notations here were confusing and we changed them. In the submission, $K_x$ and $K_y$  as defined in l207 refer to standard  Gram matrices related to the kernel $k$ and the sample set $x_1,\dots, x_n$ or $y_1,\dots, y_m $. The matrix $K_{xy}$ is the Gram matrix related to the two sample sets. These notations hence refer to Gram matrices and not covariance operators. In a nutshell, our statement in Proposition 5 shows that the regularized KKL for two empirical distributions $\hat{p},\hat{q}$ writes as a simple function of Gram kernel matrices.
> \vspace{2mm}
>
>
> - Minor typos: thanks for spotting these, we corrected accordingly.

---

### Official Review · Reviewer_r7ey · 2024-07-24

**Soundness:** 3
**Presentation:** 4
**Contribution:** 3
**Rating:** 6
**Confidence:** 5

**Summary:**

### Summary:



In this paper, the authors study kernel KL divergences (Equation 1). First, they extend the definition of kernel KL to a new setting via regularization that works for distributions with disjoint support (Equation 2).
In Propositions 2,3, they prove the approximation results for the regularized kernel KL divergence compared to the unregularized version of it.
Second, they derive convergence rates for finite sample estimation of kernel KL divergences (Proposition 3). Moreover, they obtain a closed-form expression for the new divergence (Proposition 5). They also study Wasserstein GF of the new divergence in the paper.

**Strengths:**

### Pros:


- very well-written paper
- well-motivated setting with comprehensive results

**Weaknesses:**

## Cons:

- the title does not reflect the contributions of the paper

**Questions:**

### Questions/Comments:

I recommend changing the title of the paper. The whole draft is devoted to the study of "regularized kernel KL divergence" but in the title there is no reference to being regularized.


- line 124 -- what do you mean by "if $\mathbb{R}^d$ is compact"

- line 131 -- typo "an interesting"

- how tight Proposition 4 is?

---

> ### Author Rebuttal · Authors · 2024-08-04
>
> Thank you for your valuable feedback and time. We have addressed your comments point-by-point below. Please don't hesitate to let us know if you have any further questions.
>
> - Weaknesses: "the title does not reflect the contributions of the paper"
>
>     **Reply:** We acknowledge the title does not reflect the fact that we are studying the regularized KKL and it is not clear that we are doing gradient flows on it, we will add this precision in the title.
>
> - Questions: "line 124 -- what do you mean by "if
>  is compact"
>
>  **Reply:** "$\mathbb{R}^d$ is compact" is a typo, it was originally a subset of $\mathbb{R}^d$. Thank you for this correction.
>
> - Questions: tightness of Prop 4
>
>  **Reply:** See general comment please.

---

> > ### Comment · Reviewer_r7ey · 2024-08-10
> >
> > Thank you! I appreciate the authors' response. I believe if this paper is accepted, then the title must definitely be changed. Also, I continue supporting the paper so I keep my positive score.

---

### Author Rebuttal · Authors · 2024-08-04

We sincerely thank all the reviewers for their positive comments on our paper, as well as for their relevant suggestions and questions. We adress here the general comments and questions.  If we have adequately addressed the reviewer's concerns, a re-evaluation would be greatly appreciated. For any unresolved issues, we are ready to engage further.


- Contributions (to all reviewers): We propose a kernel-based divergence relying on kernel covariance operators (in contrast to kernel mean embeddings operators as used in the Maximum Mean Discrepancy)
that is computable in closed-form (see our Proposition 5). We derive theoretical results on this divergence, including monotony with respect to the regularization parameter, statistical rates, and well-posedness and closed-form for its derivatives. This enabled us to use the latter object in a gradient flow setting to study its empirical behavior when approximating probability distributions and performs better than the MMD and is more practical than alternative kernel-based approximations of the Kullback-Leibler divergence such as KALE (which is not closed-form).
We think our study motivates that using higher order moments (eg covariances instead of mean embeddings) enables to compare probability distributions in a more powerful manner than Maximum Mean Discrepancy, while being closed-form and tractable.

- About the assumption $p\ll q$ in Proposition 2 (reviewer 19Wg and  6zq3) : This is a typo and we thank the reviewers for spotting this incoherence. This condition is not necessary to ensure the decreasing property of the regularized KKL in $\alpha$ for $\alpha \in ]0,1[$, as can be seen in its proof in Section  B.1, as $KKL_{\alpha}$ can be defined even without $p \ll q$.

- About the bound in Proposition 4 and its tightness (reviewers 19Wg and r7ey), we simplified our bound which yields the following : $$ \mathbb{E} | KKL_{\alpha}(\hat{p}||\hat{q}) - KKL_{\alpha}(p||q) | \leqslant \frac{35}{\sqrt{m \wedge n}} \frac1{\alpha \mu} (2 \sqrt{c} + \log n) + \frac{1}{m \wedge n} \left(1+\frac1{\mu} +   \frac{c (24 \log n)^2}{\alpha \mu^2} (1 + \frac{n}{m \wedge m})\right).$$ This makes it easier to see which terms dominate the upper bound. And we see, as mentioned on line 184, that if we put $n=m$, we find a bound similar to that of Proposition 7 of [1].  Notice that this is a similar rate (up to log factors) as the one of MMD that is minimax [2] and that rely on the estimation of similar Bochner integrals (kernel mean embeddings instead of kernel covariance operators).
We did not tackle the study of lower bounds, as the calculation of our upper bound was already very technical (see the sketch of proof line 190, full proof is deferred to the appendix); we leave the question of lower bound for future work.

- About Figure 2.a (reviewer 6zq3), we have attached a pdf containing the corrected version of Figure 2.a.

[1] Bach, F. Information Theory with Kernel Methods, IEEE Transactions in Information Theory,  2023.

[2] Tolstikhin, I., Sriperumbudur, B. K.,  Mu, K.  Minimax Estimation of Kernel Mean Embeddings, JMLR, 2017.

---

### Decision · Program_Chairs · 2024-09-25

**Decision:**

Accept (poster)

**Comment:**

The paper proposes a regularized version of the Kernel Kullback-Leibler divergence (KKL, Bach 2022) and studies its statistical properties, including in particular a finite-sample bound and the associated Wasserstein gradient flow.

Reviewers all agree that the presented results are novel and of interest to the community.

Additional comments:

- The title should be changed, as suggested by Reviewer r7ey.

- Eq(3): the second term on the right hand side should read $\Sigma_p(\Sigma_q + \beta I)^{-1}$. The exact reference for this identity in Ando (1979), i.e. which equation, page in that paper, should be given.

- Proposition 3: as mentioned by Reviewer  6zq3, this bound is trivial when $\alpha \rightarrow 1$, so it's a loose bound.

- In Proposition 4, the assumption that $p << q$ is a weakness, since the regularized KKL is supposed to be finite for all pairs p,q. One should be able to obtain finite-sample complexity in this general setting.